# Characterisation of the Complete Chloroplast Genomes of Seven *Hyacinthus orientalis* L. Cultivars: Insights into Cultivar Phylogeny

**Kwan-Ho Wong** [1,2], **Hoi-Yan Wu** [3], **Bobby Lim-Ho Kong** [2,3], **Grace Wing-Chiu But** [2], **Tin-Yan Siu** [1,4], **Jerome Ho-Lam Hui** [2,*], **Pang-Chui Shaw** [2,3,5,*] and **David Tai-Wai Lau** [1,3,*]

1. Shiu-Ying Hu Herbarium, School of Life Sciences, The Chinese University of Hong Kong, Shatin, Hong Kong, China; kwanhowong@cuhk.edu.hk (K.-H.W.); joycesiuty@hotmail.com (T.-Y.S.)
2. School of Life Sciences, The Chinese University of Hong Kong, Shatin, Hong Kong, China; konglimho@yahoo.com.hk (B.L.-H.K.); gracebut@link.cuhk.edu.hk (G.W.-C.B.)
3. Li Dak Sum Yip Yio Chin R & D Centre for Chinese Medicine, The Chinese University of Hong Kong, Shatin, Hong Kong, China; karenwhy@cuhk.edu.hk
4. School of Biological Sciences, The University of Hong Kong, Pokfulam, Hong Kong, China
5. State Key Laboratory of Research on Bioactivities and Clinical Applications of Medicinal Plants (The Chinese University of Hong Kong), Institute of Chinese Medicine, The Chinese University of Hong Kong, Shatin, Hong Kong, China
* Correspondence: jeromehui@cuhk.edu.hk (J.H.-L.H.); pcshaw@cuhk.edu.hk (P.-C.S.); lautaiwai@cuhk.edu.hk (D.T.-W.L.)

**Abstract:** To improve agricultural performance and obtain potential economic benefits, an understanding of phylogenetic relationships of *Hyacinthus* cultivars is needed. This study aims to revisit the phylogenetic relationships of *Hyacinthus* cultivars using complete chloroplast genomes. Nine chloroplast genomes were de novo sequenced, assembled and annotated from seven cultivars of *Hyacinthus orientalis* and two Scilloideae species including *Bellevalia paradoxa* and *Scilla siberica*. The chloroplast genomes of *Hyacinthus* cultivars ranged from 154,458 bp to 154,641 bp, while those of *Bellevalia paradoxa* and *Scilla siberica* were 154,020 bp and 154,943 bp, respectively. Each chloroplast genome was annotated with 133 genes, including 87 protein-coding genes, 38 transfer RNA genes and 8 ribosomal RNA genes. Simple sequence repeats AAGC/CTTG and ACTAT/AGTAT were identified only in 'Eros', while AAATC/ATTTG were identified in all cultivars except 'Eros'. Five haplotypes were identified based on 460 variable sites. Combined with six other previously published chloroplast genomes of Scilloideae, a sliding window analysis and a phylogenetic analysis were performed. Divergence hotspots *ndhA* and *trnG-UGC* were identified with a nucleotide diversity threshold at 0.04. The phylogenetic positions of *Hyacinthus* cultivars were different from the previous study using ISSR. Complete chloroplast genomes serve as new evidence in *Hyacinthus* cultivar phylogeny, contributing to cultivar identification, preservation and breeding.

**Keywords:** *Hyacinthus orientalis*; hyacinth; *Scilla siberica*; *Bellevalia paraxoda*; chloroplast genome; cultivar phylogeny; geophytes; Asparagaceae; Scilloideae; Hyacinthaceae

## 1. Introduction

### 1.1. Taxonomy of Hyacinthus orientalis L.

#### 1.1.1. Morphology

*Hyacinthus orientalis* L., commonly known as hyacinth, is one of the most important cultivated plants around the world [1–3]. The cultivars of this species are characterised by their flowers with strong fragrances [1,4–9] and a wide range of attractive colours [1,4,5,7,10,11], which make them a beloved option among ornamentals [7–9].

As a geophyte [3,12], the hyacinth has a significant, globose bulb [5] (Figure 1A,E) which is modified from its stem and leaves [13,14]. Its stem is shrunken and flattened as the

disc (Figure 1F), while the modified leaves become scale leaves [13,14] (Figure S1) acting as the storage of nutrients [7]. The bulb of the hyacinth is tunicate [4,5], meaning that the outermost layers of scale leaves turn into a thin and dry cover called a tunic to protect the inner, fresh scale leaves [13,15]. The outer tunics of the hyacinth show different colours depending on cultivars [5], which can be generally classified into three major colours: dark purple, beige-to-white, and silvery purple.

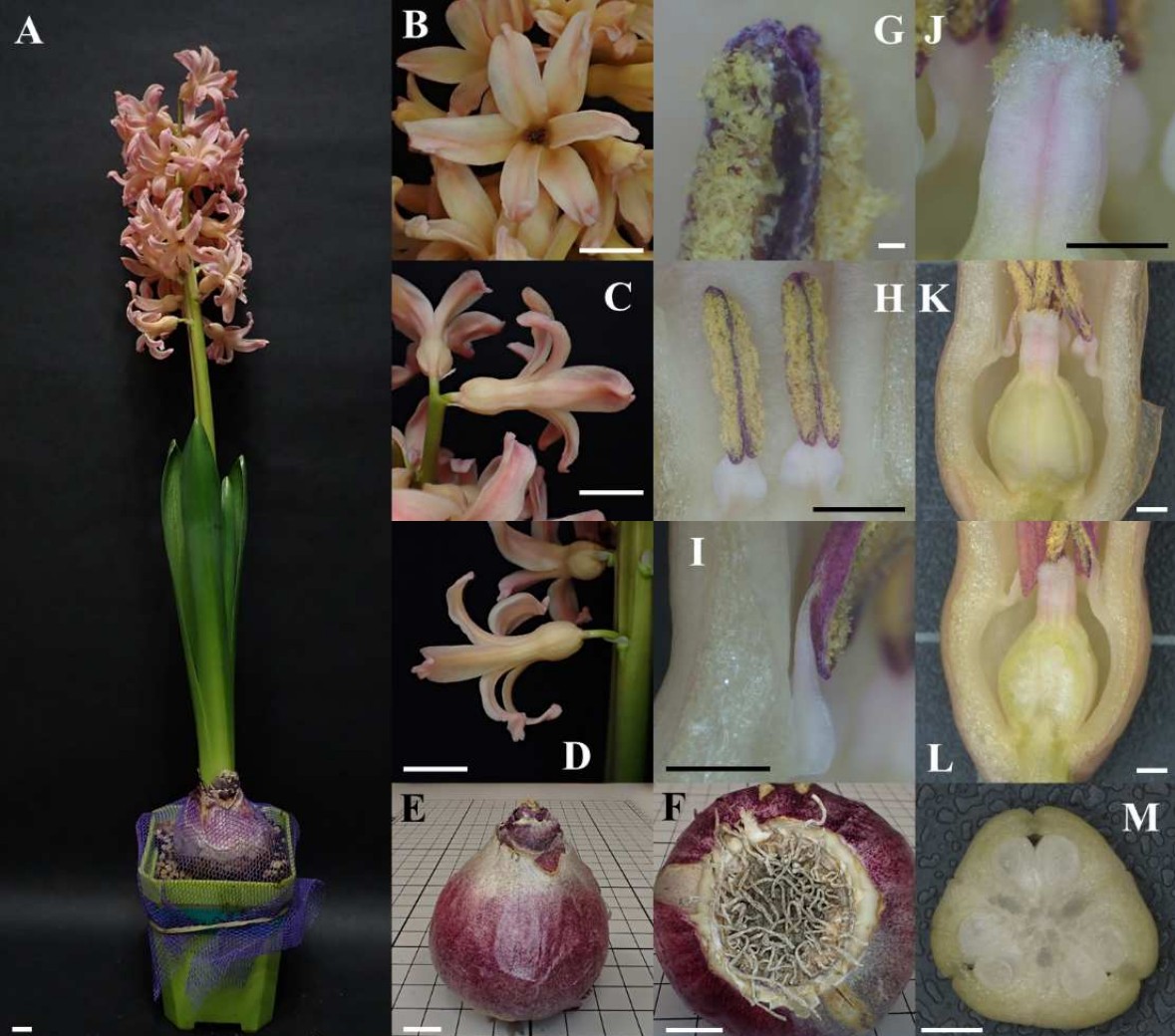

**Figure 1.** Overview and close-up photos of *Hyacinthus orientalis* L. 'Gipsy Queen'. (**A**) Overview of an individual of *Hyacinthus orientalis* L. 'Gipsy Queen' (scale bar = 1 cm). (**B–D**) The corolla (scale bar = 1 cm). (**B**) Front view. (**C**) Side view of corolla on the upper inflorescence. (**D**) Side view of corolla on the lower inflorescence. (**E,F**) The bulb (scale bar = 1 cm). (**E**) Side view of the bulb of which tunic is in slivery purple. (**F**) The disc with remnant roots. (**G–I**) Androecium. (**G**) Pollen sacs under microscopy (scale bar = 0.01 cm). (**H**) Stamens (scale bar = 0.1 cm). (**I**) Filaments of dorsifixed anthers (scale bar = 0.1 cm). (**J–M**) Gynoecium (scale bar = 0.1 cm). (**J**) Receptive papilla on the stigma. (**K**) Superior ovary. (**L**) Longitudinal section of the ovary showing axile placentation. (**M**) Cross section of the ovary showing ovules in three locules.

The inflorescence of the hyacinth is a raceme with 2 to 40 flowers on a single scape [1,5] (leafless stalk arising from ground level [13,15]). The number and density of flowers vary across cultivars. Wild populations of *Hyacinthus orientalis* blossom in comparatively looser scapes, bearing fewer flowers in a blue colour [5,16]. Cultivars of hyacinths have flowers in different colours [1,4–6,8–10,17–19], including red, pink, orange, yellow, white, blue

and purple. The flower normally has six tepals arranged in two whorls ($P_{3+3}$) (Figure 1B), which are basally united forming a perianth tube (Figure 1C,D) and constricted above the trilocular superior ovary ($\underline{G}_{(3)}$) [5] with axile placentation (Figure 1L,M). The perianth lobes are oblong–spathulate in shape, spreading to recurved [5,10]. Stamens of the hyacinth are in the same number as tepals ($A_{3+3}$) and attached to the tepals while not reaching to the throat of the perianth tube [5]. The linear, longitudinally dehiscent dorsifixed anthers are longer than the filaments [5] (Figure 1H,I) and show different colours depending on the cultivars [19]. Different cultivars of hyacinths can either blossom with single or double flowers [16,18–20].

Hyacinths originate from the Mediterranean region, including Turkey, Syria and Lebanon [1,5,10,19,21]. In Turkey, the species naturally inhabits rocky limestone slopes and scrubs [21]. The species is also found in Israel where it appears in small stands under the shade of maquis among rocks [12,22].

### 1.1.2. Nomenclature

The scientific name *Hyacinthus orientalis* L. was first published by Carl Linnaeus in 1753 in *Species Plantarum* [23]. The species has an indispensable role in the nomenclature of related taxa. Being the type species of the genus *Hyacinthus* L., *Hyacinthus orientalis* L. also represented the family Hyacinthaceae Batsch ex Borkh. as *Hyacinthus* L. is the designated type genus [24,25]. The family was proposed by Borkhausen in 1797 [26] and has been recognised by several taxonomists and taxonomic systems including Dahlgren et al. in 1985 [27], Conran et al. in 2005 [10], the NCBI Taxonomy system [28] and the Angiosperm Phylogeny Group (APG) II system in 2003 [29]. However, since 2009, the APG system has adopted broader limits for organising the families in Asparagales [30,31]. The family Hyacinthaceae was included in the family Asparagaceae *sensu lato* and ranked as a subfamily [30–32]. According to the International Code for Nomenclature of Plants, Algae and Fungi (ICN), no conserved names are allowed for taxa below family level [33]. Since the subfamily name Hyacinthoideae (Link, 1829) [34] had been previously adopted, according to the priority of names in ICN, the subfamily name Scilloideae (Burnett, 1835) [35] was adopted for such inclusion [32].

### 1.2. Cultivation of Hyacinthus orientalis L.

#### 1.2.1. History of Cultivation

The cultivation of hyacinth is long-standing with 460 years of history. The earliest record can be traced back to 1562, when the hyacinth was imported from Turkey to Eastern Europe [20]. At the end of sixteenth century, the hyacinth was introduced into England as a cultivated plant [36]. Cultivars of hyacinths were produced through either hybridisation or mutation [7,11]. The climax of hyacinth cultivation should be dated back to the 1760s, when over 2000 cultivars of hyacinths were recorded in the French Monograph *Des Jacintes de Leur Anatomie Reproduction et Culture* published by Saint Simon in 1768 [14]. However, most of these cultivars were unable to survive [20].

#### 1.2.2. Nomenclatural Circumscription of Hyacinthus Cultivars

Homonyms and synonyms of cultivar epithets are serious issues in the nomenclature of *Hyacinthus* cultivars [8,11,16,20]. Distinct cultivars have shared an identical cultivar epithet, causing the problem of homonyms. For example, three cultivars with single blue flowers from three different origins—Haarlem, Overveen and Hillegom—shared the epithet 'Queen of the Blues' [16]. Another cultivar epithet 'Grand Vainqueur' was severely abused, as almost all colours of flowers, regardless of single or double, were called this epithet [16]. In contrast, a single cultivar was given two or more cultivar epithets, causing the problem of synonyms. For example, the registered 'Orange Boven' was given the unaccepted epithet 'Salmonetta' [20]; the registered 'China Pink' was misapplied with the epithet 'Delft Pink' [19]; the registered 'Kroonprinses Margaretha' was given two misapplied epithets, 'Crownprincess Margareth' and 'Margareth' [19].

Since 1955, Koninklijke Algemeene Vereeniging voor Bloembollencultuur (KAVB) in the Netherlands was appointed as an International Cultivar Registration Authority (ICRA) for hyacinths by the International Society of Horticulture (ISHS) Commission for Nomenclature and Cultivar Registration [37,38]. In the *International Checklist for Hyacinths and Miscellaneous Bulbs* published by KAVB in 1991, there were 202 registered cultivars of hyacinths [19]. In 1993, about 50 cultivars were commercialised in floricultural production [7]. As of 1 January 2020, 368 registered cultivars of hyacinths were recorded in the database of KAVB [39]. According to the *International Code of Nomenclature for Cultivated Plants* (ICNCP), for the plants governed by ICRA, each cultivar can only be given one accepted name [38].

*1.3. Recent Molecular Insight into Hyacinthus Cultivars and the Potentiality of Chloroplast Genomes for Cultivar Phylogeny*

The compatibility of cross-fertilisation and phylogenetic relationships among the *Hyacinthus* cultivars have been studied by karyotypic [8,11,20,40–42] and molecular means [9,43], respectively. The diversity of chromosomal karyotypes was characterised in *Hyacinthus* cultivars, which can be diploid, triploid, tetraploid and aneuploid [8,20,40]. It is difficult to identify the authenticity of hybrid offspring in hyacinths ascribed by their richness of chromosomal ploidy variation and also the greater chances to obtain hybrid offspring from parents with higher ploidy [8,20,42,43]. As the hyacinth grows only in the right season and starts to blossom in the 2nd to 3rd year from seeds [2,4], the cost of breeding a new cultivar can be greatly reduced by early identification and selection [43].

The research group of Hu et al. utilised twelve Inter-Simple Sequence Repeat (ISSR) molecular markers to analyse the phylogenetic relationships of 29 *Hyacinthus* cultivars [9]. In their unweighted pair group method with arithmetic mean (UPGMA) tree, cultivars with the same colour were mostly grouped into the same cluster. They concluded that the phylogenetic relationships among *Hyacinthus* cultivars had a correlation with the flower colours to a certain extent [9]. The research group continued to identify hyacinth hybrid progeny using the twelve ISSR molecular markers [43]. The authenticity of hybrid offspring was assured by the presence of parental bands and offspring-unique bands in the electrophoresis diagram of the ISSR analysis [43].

With recent technological advancements, the assembly of complete chloroplast genomes has become more feasible. Apart from resolving phylogenetic problems [44–47], chloroplast genomes were recently applied in studying ornamental plants such as *Lilium* L. [48,49], *Camellia* L. [50], *Lagerstroemia* L. [51], *Meconopsis* Vig. [52] and *Paeonia* L. [53]. The chloroplast genomes were utilised to reconstruct the phylogenetic relationship in horticultural species [48–50] and sometimes even up to the cultivar level [51,54,55].

Currently, only six complete chloroplast genomes of Scilloideae are available, including three of *Barnardia japonica* (Thunb.) Schult. et Schult. f. (NC_035997 = KX822775, MH287351 [56] and MT319125 [57]), one of *Hyacinthoides non-scripta* (L.) Chouard ex Rothm. (NC_046498 = MN824434) [58], one of *Albuca kirkii* (Baker) Brenan (NC_032697 = KX931448) [59] and one of *Oziroe biflora* (Ruiz et Pav.) Speta (NC_032709 = KX931463) [59]. To date, this is the first report to present the chloroplast genomes of the genus *Hyacinthus* L., *Bellevalia* Lapeyr. and *Scilla* L., which were members of Scilloideae of Asparagaceae *sensu* APG IV. In total, nine chloroplast genomes were sequenced and assembled in this study using Illumina sequencing technology, providing important germplasm resources and insight for the cultivar breeding of the hyacinth and its relatives.

## 2. Materials and Methods

*2.1. Plant Material and DNA Extraction*

Bulbs of all studied specimens were imported from the Netherlands (Simple Pleasures Flowerbulbs & Perennials Inc.) in September of 2019 or 2020 (Table 1). The bulbs were immersed in the fungicide tetrachlorophthalonitrile ($C_8Cl_4N_2$) for 1 h and then air-dried before being stored at 4 °C for three months to vernalise the bulbs. After the low-temperature treatment, the bulbs were planted in an indoor environment. Whole plants were harvested

in January or February of the next year (Table 1) when the inflorescences were in anthesis (Figure 2). Healthy leaves were stored at −80 °C immediately once cut from the individuals. The remaining materials including the bulbs were processed into specimens, which were deposited in the Shiu-Ying Hu Herbarium (herbarium code: CUHK), School of Life Sciences, The Chinese University of Hong Kong, as vouchers (Table 1 and Figure S1). Freshly blossoming flowers of each cultivar were studied under stereomicroscope (SMZ745T, Nikon Instruments Inc., Melville, NY, USA), and the details including anthers, pistils and ovules, were attached in Figure S2. We authenticated the cultivars of *Hyacinthus orientalis* L. and the specimens of *Scilla siberica* Haw. based on the *International Checklist for Hyacinths and Miscellaneous Bulbs* [19], while the specimen of *Bellevalia paradoxa* (Fisch. et Mey.) Boiss. was authenticated by referring to the *Flora of Turkey*, Volume 8 [60] and the specimen Davis et Hedge 30534, E00340807 (http://data.rbge.org.uk/herb/E00340807 (accessed on 31 January 2022)).

**Table 1.** Information of the studied specimens.

| Species | Collector No. | Inventory No. | Sheet No. | Date of Collection | NCBI Accession No. | Raw Data (GB) | Coverage |
|---|---|---|---|---|---|---|---|
| *Hyacinthus orientalis* L. 'Gipsy Queen' | K. H. Wong 127 | CUSLSH2896 | CUHK05312 | 27 January 2021 | OM320803 | 3.2 | 308× |
| *Hyacinthus orientalis* L. 'City of Haarlem' | K. H. Wong 139 | CUSLSH2915 | CUHK05574 | 11 February 2021 | OM320804 | 3.5 | 203× |
| *Hyacinthus orientalis* L. 'Eros' | K. H. Wong 144 | CUSLSH2922 | CUHK05309 | 15 February 2021 | OM320805 | 4.1 | 194× |
| *Hyacinthus orientalis* L. 'Chicago' | K. H. Wong 047 | CUSLSH2460 | CUHK05305 | 22 January 2020 | OM320806 | 3.7 | 138× |
| *Hyacinthus orientalis* L. 'Woodstock' | K. H. Wong 114 | CUSLSH2881 | CUHK05317 | 22 January 2021 | OM320807 | 3.5 | 279× |
| *Hyacinthus orientalis* L. 'Delft Blue' | K. H. Wong 118 | CUSLSH2885 | CUHK05307 | 22 January 2021 | OM320809 | 4.1 | 221× |
| *Hyacinthus orientalis* L. 'Aiolos' | K. H. Wong 120 | CUSLSH2887 | CUHK05303 | 22 January 2021 | OM320808 | 3.5 | 255× |
| *Bellevalia paradoxa* (Fisch. et C.A.Mey.) Boiss. | K. H. Wong 126 | CUSLSH2893 | CUHK05320 | 24 January 2021 | OM320811 | 3.8 | 338× |
| *Scilla siberica* Haw. | K. H. Wong 141 | CUSLSH2917 | CUHK05324 | 11 February 2021 | OM320810 | 3.0 | 231× |

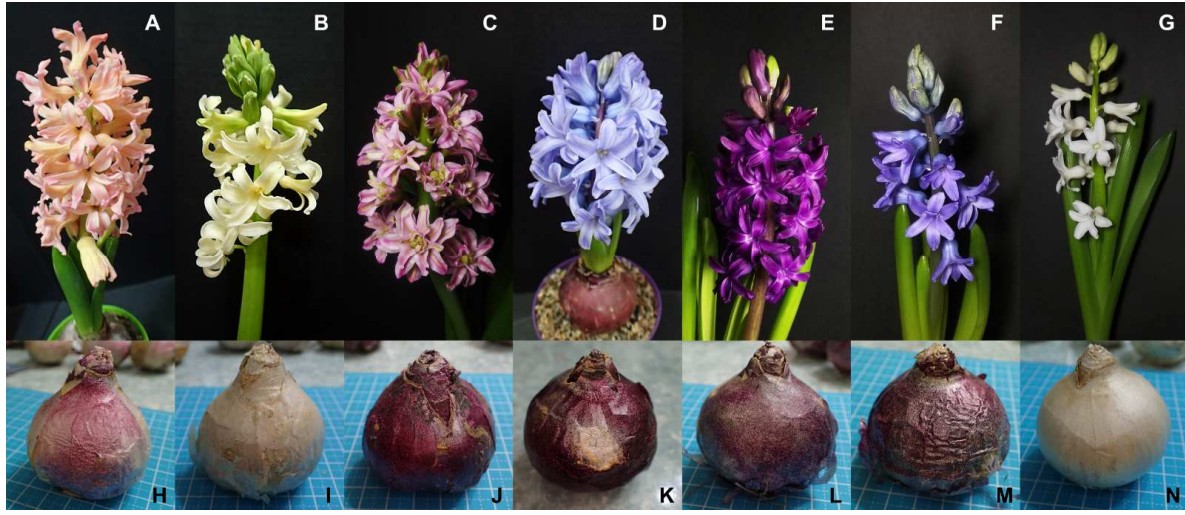

**Figure 2.** Inflorescences and tunicate bulbs of the seven *Hyacinthus* cultivars. (**A–G**) Inflorescences. (**H–N**) Tunicate bulbs. (**A,H**) 'Gipsy Queen'. (**B,I**) 'City of Haarlem'. (**C,J**) 'Eros'. (**D,K**) 'Chicago'. (**E,L**) 'Woodstocks'. (**F,M**) 'Delft Blue'. (**G,N**) 'Aiolos'.

Genomic DNA of each specimen was extracted from about 50 mg of frozen fresh leaves using i-genomic Plant DNA Extraction Mini Kit (iNtRON Biotechnology, Daejeon, Korea) according to the instructions of the manufacturer. The concentration of extracted DNA

was measured by NanoDrop Lite Spectrophotometer (Thermo Fisher Scientific, Waltham, MA, USA), while the quality of DNA was checked by 1% agarose gel electrophoresis. The qualified genomic DNA was sent to Novogene Bioinformatic Technology Co. Ltd. (http://en.novogene.com/ (accessed on 31 January 2022), Beijing, China) for shotgun sequencing.

### 2.2. Chloroplast Genome Sequencing, Assembly and Annotation

A paired-end library with an insert-size of 150 bp was constructed and sequenced on the NovaSeq 6000 platform (Illumina Inc., San Diego, CA, USA). CLC Genomic Workbench version 21.0.4 (CLC Inc., Aarhus, Denmark) was used for the chloroplast genome assembly. About 3 GB raw reads of each specimen were firstly paired up and quality trimmed, then de novo assembled into contigs. Contigs in the length of 1000 bp or longer were mapped against the reference chloroplast genome, *Hyacinthoides non-scripta* (NC_046498), which was selected based on its phylogenetic proximity. Visualisation of the alignment was achieved by Comparative Viewer (http://phyzen.iptime.org/tools/cv.php (accessed on 31 January 2022), Phyzen, Seongnam, Korea). After correcting the directionality (either 5′ to 3′ or 3′ to 5′) of the candidate contigs, the overlapping sequences between these contigs were trimmed. Trimmed contigs were merged into a single sequence as a chloroplast genome, after ensuring there were no undetermined "N" nucleotides.

Gene annotation of chloroplast genome was performed on the GeSeq platform (https://chlorobox.mpimp-golm.mpg.de/geseq.html (accessed on 31 January 2022)) [61] based on the NCBI-verified chloroplast genomes *Hyacinthoides non-scripta* (NC_046498) and *Barnardia japonica* (NC_035997). Manual adjustments, including editing the start and stop positions for the genes and introns, were performed when necessary. Visualisation of the circular genomic maps was performed using OrganellarGenomeDRAW (OGDRAW, https://chlorobox.mpimp-golm.mpg.de/OGDraw.html (accessed on 31 January 2022)) [62]. The assembled and annotated chloroplast genomes were then submitted to the GenBank with the accession number OM320803 to OM320811 (Table 1).

### 2.3. Sequence Repeat Analysis

In addition to conducting a comparative analysis of the chloroplast genomes of the seven *Hyacinthus* cultivars, the chloroplast genomes of the two Scilloideae species *Scilla siberica* and *Bellevalia paradoxa* were also investigated.

MIcroSAtellite identification tool (MISA, https://webblast.ipk-gatersleben.de/misa/index.php?action=1 (accessed on 31 January 2022)) [63] was used to identify Simple Sequence Repeats (SSRs) from the nine chloroplast genomes. The minimum number of repetitions for mono-, di, tri-, tetra-, penta- and hexa-nucleotides was set to 10, 5, 4, 3, 3 and 3, respectively.

REPuter (https://bibiserv.cebitec.uni-bielefeld.de/reputer (accessed on 31 January 2022)) [64] was applied to detect Long Tandem Repeats (LTRs), including forward, reverse, complement and palindromic sequences. The maximum computed repeat size and minimal repeat size were set to 50 bp and 30 bp, respectively.

### 2.4. Visualisation of the Boundary Variations

The visualisation diagram of the junctions within the nine chloroplast genomes was drawn manually based on the results of annotation by checking the size and position of the border-flanking genes, as well as the length of LSC, SSC and IRs in each chloroplast genome.

### 2.5. Comparative and Phylogenetic Analysis

All Scilloideae chloroplast genomes available in the NCBI GenBank (https://www.ncbi.nlm.nih.gov/genbank/ (accessed on 31 January 2022)) (NC_035997, MH287351, MT319125, NC_046498, NC_032697 and NC_032709), together with the nine newly assembled chloroplast genomes for this study, were comparatively analysed.

mVISTA (https://genome.lbl.gov/vista/mvista/submit.shtml (accessed on 31 January 2022)) [65] was adopted to visualise the structural variation of the fifteen chloroplast genomes in full alignment, using the chloroplast genome of *Hyacinthus orientalis* L. 'Gipsy Queen' as a reference. Shuffle-LAGAN alignment programme [66] was selected.

MAFFT version 7 (https://mafft.cbrc.jp/alignment/server/ (accessed on 31 January 2022)) [67] was applied to align chloroplast genomes. Cases of alignment included (i) seven *Hyacinthus* cultivars, (ii) seven *Hyacinthus* cultivars + *Scilla siberica* + *Bellevalia paradoxa* and (iii) seven *Hyacinthus* cultivars + *Scilla siberica* + *Bellevalia paradoxa* + *Hyacinthoides non-scripta* + three *Barnardia japonica* + *Albuca kirkii* + *Oziroe biflora*.

DNA Sequence Polymorphism (DnaSP) version 6.12.03 [68] was adopted to calculate the nucleotide diversity values (Pi) from the aligned chloroplast genomes in the three cases above. The window length was set to 600 bp, and the step size was set to 200 bp.

Maximum Likelihood (ML) tree was constructed using the software MEGA-X version 10.2.5 [69]. The best-fit nucleotide substitution model with the lowest Bayesian Information Criterion (BIC) scores was selected. The bootstrap replicates were set to 1000. The full alignment of the fifteen complete chloroplast genomes and the loci with high nucleotide diversity were used for the construction of ML trees.

### 2.6. Haplotype Grouping and Identification of Molecular Diagnostic Characters

To increase the resolution of the differences among the seven chloroplast genomes of *Hyacinthus* cultivars, haplotype grouping was performed. The haplotype data file was generated using DnaSP [68] considering the alignment gaps. The haplotype data file was used to compute a median joining network [70] using Network version 10.2.0.0 [71]. Variation sites, including Single-Nucleotide Polymorphisms (SNPs) and Insertions–deletions (Indels), were then manually identified in the alignment using BioEdit [72]. The Molecular Diagnostic Characters (MDCs) were then manually identified from SNPs and Indels.

## 3. Results

### 3.1. Features of Hyacinthus Chloroplast Genomes

3.1.1. Genome Size

The chloroplast genomes of the seven studied *Hyacinthus* cultivars were highly conserved. Regarding the size of the chloroplast genomes, the largest chloroplast genome was *Hyacinthus* 'Chicago' with 154,641 bp. The chloroplast genomes of three *Hyacinthus* cultivars, i.e., 'Woodstocks', 'Delft Blue' and 'Aiolos', were identical with 154,640 bp, which were just one base pair smaller than 'Chicago' (Table 2). The smallest chloroplast genome was the one of the double-flower cultivar 'Eros' with 154,458 bp, which was smaller than the biggest one by 183 bp.

All newly assembled chloroplast genomes in this study demonstrated a quadripartite structure of chloroplast genomes typical of angiosperms of which a Large Single Copy (LSC) and a Small Single Copy (SSC) are separated by a pair of Inverted Repeat (IR) regions (Figures S3 and 3). The chloroplast genomes of four cultivars, namely 'Chicago', 'Delft Blue', 'Woodstock' and 'Aiolos', had the same size of LSC and IR, which were 83,159 bp and 26,503 bp, respectively. The one extra base pair from the chloroplast genome of 'Chicago', when compared to 'Delft Blue', 'Woodstock' and 'Aiolos', was found in the SSC region (Table 2).

**Table 2.** Summary of the chloroplast genome structures of the studied specimens.

| | *Hyacinthus orientalis* 'Gipsy Queen' | *Hyacinthus orientalis* 'City of Haarlem' | *Hyacinthus orientalis* 'Eros' | *Hyacinthus orientalis* 'Chicago' | *Hyacinthus orientalis* 'Wood-stock' | *Hyacinthus orientalis* 'Delft Blue' | *Hyacinthus orientalis* 'Aiolos' | *Bellevalia para-doxa* | *Scilla siberica* |
|---|---|---|---|---|---|---|---|---|---|
| Total length (bp) | 154,630 | 154,599 | 154,458 | 154,641 | 154,640 | 154,640 | 154,640 | 154,020 | 154,943 |
| LSC (bp) | 83,180 | 83,149 | 83,168 | 83,159 | 83,159 | 83,159 | 83,159 | 83,028 | 83,384 |
| SSC (bp) | 18,462 | 18,462 | 18,302 | 18,476 | 18,475 | 18,475 | 18,475 | 18,570 | 18,449 |
| IR (bp) | 26,494 | 26,494 | 26,494 | 26,503 | 26,503 | 26,503 | 26,503 | 26,211 | 26,555 |
| Gene no. | 133 | 133 | 133 | 133 | 133 | 133 | 133 | 133 | 133 |
| mRNA | 87 | 87 | 87 | 87 | 87 | 87 | 87 | 87 | 87 |
| tRNA | 38 | 38 | 38 | 38 | 38 | 38 | 38 | 38 | 38 |
| rRNA | 8 | 8 | 8 | 8 | 8 | 8 | 8 | 8 | 8 |
| Pseudogene (Ψ) | 1 | 1 | 1 | 1 | 1 | 1 | 1 | 1 | 1 |
| 1-intron gene | 21 | 21 | 21 | 21 | 21 | 21 | 21 | 21 | 21 |
| 2-introns gene | 2 | 2 | 2 | 2 | 2 | 2 | 2 | 2 | 2 |
| GC content (%) | 37.58 | 37.58 | 37.60 | 37.58 | 37.58 | 37.58 | 37.58 | 37.66 | 37.60 |
| A content (%) | 30.86 | 30.87 | 30.84 | 30.86 | 30.86 | 30.86 | 30.86 | 30.83 | 30.85 |
| C content (%) | 19.13 | 19.13 | 19.14 | 19.13 | 19.13 | 19.13 | 19.13 | 19.16 | 19.14 |
| G content (%) | 18.45 | 18.45 | 18.46 | 18.45 | 18.45 | 18.45 | 18.45 | 18.50 | 18.46 |
| T content (%) | 31.55 | 31.55 | 31.55 | 31.56 | 31.56 | 31.56 | 31.56 | 31.51 | 31.55 |

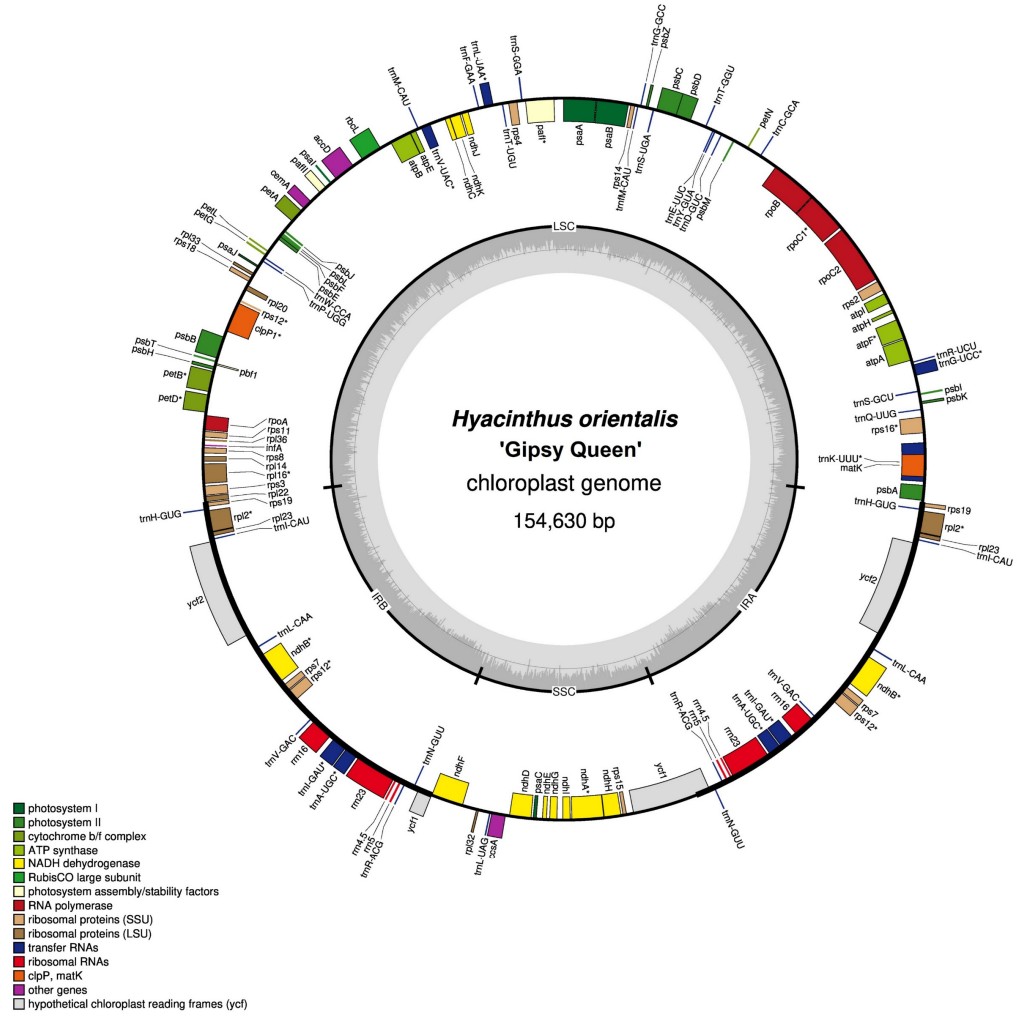

**Figure 3.** Chloroplast genome map of *Hyacinthus orientalis* L. 'Gipsy Queen'. Intron-containing genes were asterisked (*).

The largest LSC was found in the chloroplast genomes of the orange-flower cultivar 'Gipsy Queen', which was 83,180 bp. In contrast, the smallest LSC was found in the yellow-flower 'City of Haarlem', which was 83,149 bp. Interestingly, the chloroplast genomes of these two cultivars shared the same size of SSC with 18,462 bp and of IR with 26,494 bp.

The different sizes of LSC resulted in the size differentiation of the chloroplast genomes in these two cultivars. The double-flower cultivar 'Eros' also had an IR of 26,494 bp. However, its SSC of 18,302 bp was the smallest among the seven cultivars, and the contraction of SSC resulted in the smallest chloroplast genome of 'Eros' among the seven.

### 3.1.2. Gene Number and Content

The gene number and content were identical across all newly assembled chloroplast genomes. In total, 133 genes were successfully annotated for each chloroplast genome, including 87 protein-coding (mRNA) genes, 38 transfer RNA (tRNA) genes and 8 ribosomal RNA (rRNA) genes. All of them had the pseudogene *ycf1* in IR$_B$ and the tran-splicing gene *rps12* of which an exon was in LSC, while two complementarily inverted exons were in IR$_B$ and IR$_A$, respectively (Figure 3 and Figure S3). Eighteen genes were duplicated as they were in the IR regions. Twenty-one genes had one intron, while the genes *clpP1* and *pafI* had two introns (Tables 2 and 3).

**Table 3.** Genes annotated in the chloroplast genomes of the studied nine specimens.

| Gene Category | Gene Functions | Gene Names |
|---|---|---|
| Photosynthesis-related genes | Rubisco | *rbcL* |
| | Photosystem I | *psaA, psaB, psaC, psaI, psaJ* |
| | Assembly/ stability of photosystem I | *pafI \*\*, pafII, pbf1* |
| | Photosystem II | *psbA, psbB, psbC, psbD, psbE, psbF, psbH, psbI, psbJ, psbK, psbL, psbM, psbT, psbZ* |
| | ATP synthase | *atpA, atpB, atpE, atpF \*, atpH, atpI* |
| | Cytochrome b/f complex | *petA, petB \*, petD \*, petG, petL, petN* |
| | Cytochrome c synthesis | *ccsA* |
| | NADPH dehydrogenase | *ndhA \*, ndhB \* (×2), ndhC, ndhD, ndhE, ndhF, ndhG, ndhH, ndhI, ndhJ, ndhK* |
| Transcription- and translation-related genes | Transcription | *rpoA, rpoB, rpoC1 \*, rpoC2* |
| | Ribosomal protein | *rpl2 \* (×2), rpl14, rpl16 \*, rpl20, rpl22, rpl23 (×2), rpl32, rpl33, rpl36, rps2, rps3, rps4, rps7 (×2), rps8, rps11, rps12 \*\* (×2, tran-spliced), rps14, rps15, rps16 \*, rps18, rps19 (×2)* |
| | Translation initiation factor | *infA* |
| RNA genes | Ribosomal RNA | *rrn4.5 (×2), rrn5 (×2), rrn16 (×2), rrn23 (×2)* |
| | Transfer RNA | *trnA-UGC \* (×2), trnC-GCA, trnD-GUC, trnE-UUC, trnF-GAA, trnfM-CAU, trnG-GCC, trnG-UCC \*, trnH-GUG (×2), trnI-CAU (×2), trnI-GAU \* (×2), trnK-UUU \*, trnL-CAA (×2), trnL-UAA \*, trnL-UAG, trnM-CAU, trnN-GUU (×2), trnP-UGG, trnQ-UUG, trnR-ACG (×2), trnR-UCU, trnS-GCU, trnS-GGA, trnS-UGA, trnT-GGU, trnT-UGU, trnV-GAC (×2), trnV-UAC \*, trnW-CCA, trnY-GUA* |
| Miscellaneous group | Maturase | *matK* |
| | Inner membrane protein | *cemA* |
| | ATP-dependent protease | *clpP1 \*\** |
| | Acetyl-CoA carboxylase | *accD* |
| | Unknown functions | *ycf1 (×2), ycf2 (×2)* |

Number of Asterisks (*) indicates the number of introns contained in the respective genes.

The genes were categorised into three major groups, namely photosynthesis-related genes, transcription- and translation-related genes and RNA genes (Table 3). The genes *ycf1* and *ycf2* as the genes of the hypothetical open reading frame (ORF) proteins were annotated across the nine chloroplast genomes.

### 3.1.3. Guanine–Cytosine (GC) Content

Regarding the composition of the nucleotides, the chloroplast genomes of the seven *Hyacinthus* cultivars were highly congruent. The total GC content of the chloroplast genomes was 37.58%, except for the one of the double-flower cultivar 'Eros', which was only 0.02% higher (Table 2). This pattern was also observed in the total cytosine (C) content; six chloroplast genomes were 19.13%, but the chloroplast genome of 'Eros' was 0.01% higher. The total Adenine (A) content varied from 30.84% to 30.87% across the chloroplast genomes of the *Hyacinthus* cultivars. The total Guanine (G) and total Thymine (T) contents only had 0.01% differences, respectively, among the chloroplast genomes of the *Hyacinthus* cultivars.

### 3.2. Sequence Repeats

#### 3.2.1. Simple Sequence Repeats (SSRs)

Regarding the repeat types of SSRs, all the nine newly assembled chloroplast genomes had mono-, di-, tri-, tetra- and penta-nucleotides SSRs (Figure 4). The abundance of different types of SSRs was almost the same across the chloroplast genomes of the seven *Hyacinthus* cultivars, except for the mononucleotide SSRs. Forty mononucleotide SSRs were identified in the chloroplast genome of the double-flower cultivar 'Eros', which was lower than other *Hyacinthus* chloroplast genomes by one.

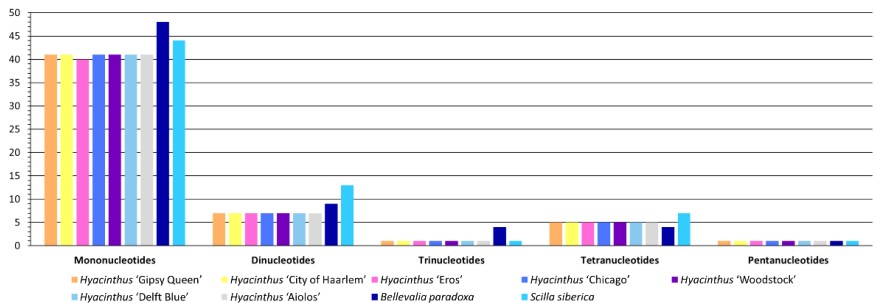

**Figure 4.** Abundance of SSRs classified by different repeat types among the nine studied chloroplast genomes.

Regarding the sequence complementarity of SSRs, eight were detected in all the chloroplast genomes of the studied *Hyacinthus* cultivars, including A/T, AG/CT, AT/AT, AAT/ATT, AAAG/CTTT, AAAT/ATTT, AATG/ATTC and AGAT/ATCT (Figure 5). It is noteworthy that only the chloroplast genome of the 'Eros' had the SSRs AAGC/CTTG and ACTAT/AGTAT. These two SSRs could be potential markers for authenticating this cultivar, in the absence of inflorescence. Meanwhile, the chloroplast genome of 'Eros' did not contain the complementary AAATC/ATTTG that the chloroplast genomes of other *Hyacinthus* cultivars did. The chloroplast genome of 'Eros' had ten complementary AT/AT, which was significantly higher when compared with the chloroplast genomes of other *Hyacinthus* cultivars; each had only four such SSRs. Additionally, the chloroplast genomes of the orange-flower cultivar 'Gipsy Queen' and the yellow-flower cultivar 'City of Haarlem' did not have the complementary C/G in contrast to the chloroplast genomes of other *Hyacinthus* cultivars where each had two such SSRs.

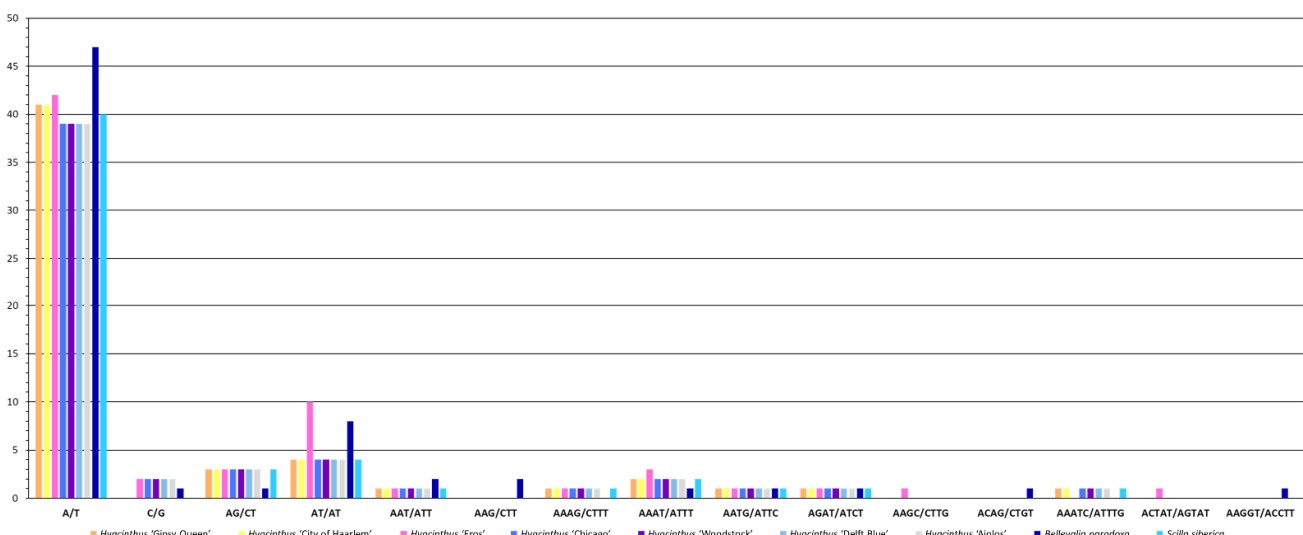

**Figure 5.** Abundance of SSRs classified by different sequence complementarity among the nine studied chloroplast genomes.

Regarding the distribution of SSRs in the partitions of the chloroplast genomes, similarity was observed across the chloroplast genomes of the studied *Hyacinthus* cultivars. All seven chloroplast genomes of the *Hyacinthus* cultivars had six SSRs in their IR regions (Figure 6). However, differences were observed in LSC and SSC. The LSC of all seven *Hyacinthus* chloroplast genomes had thirty-eight or thirty-nine SSRs. The number of SSRs in the SSC of *Hyacinthus* chloroplast genomes except for the one of 'Eros' was either seven or eight. 'Eros' again contained the lowest abundance of SSRs, i.e., six in SSC.

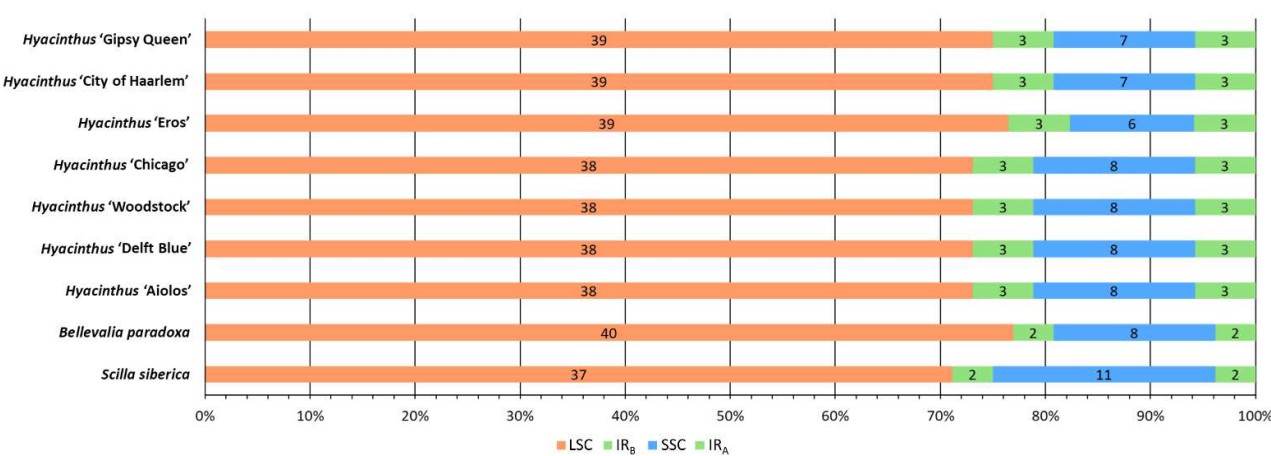

**Figure 6.** Distribution of SSRs in different partitions of the nine studied chloroplast genomes. The partitions of chloroplast genome were abbreviated as the following: Large Single Copy—LSC; IR_B—Inverted Repeat B; Small Single Copy—SSC; IR_A—Inverted Repeat A.

### 3.2.2. Long Tandem Repeats (LTRs)

Only two types of LTRs, forward repeats and palindromic repeats, were detected in the nine newly assembled chloroplast genomes (Figure 7). The number of forward repeats was the same across the chloroplast genomes of *Hyacinthus*, i.e., two. The chloroplast genome of *Bellevalia paradoxa* had one more forward repeat than the other newly assembled chloroplast genomes. The chloroplast genome of 'City of Haarlem' and *Scilla siberica* had one less palindromic repeat than the other studied chloroplast genomes.

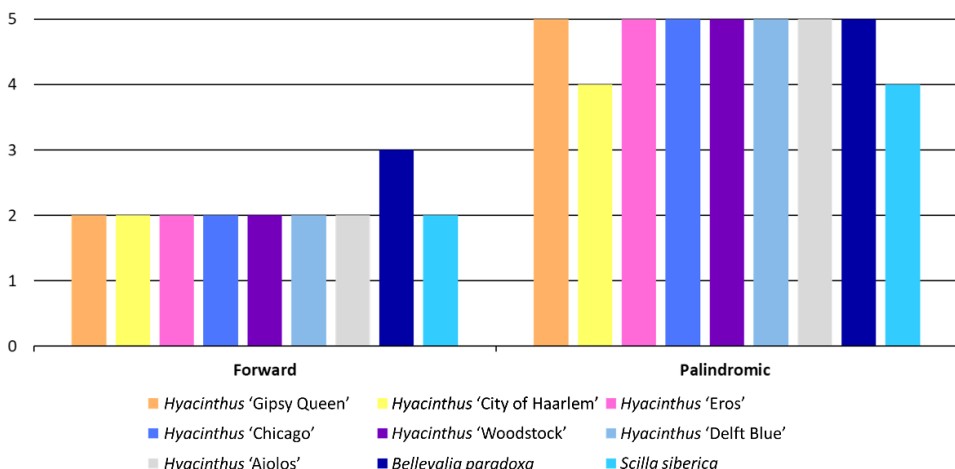

**Figure 7.** Abundance of LTRs by types among the nine studied chloroplast genomes.

Regarding the length of the LTRs, one 32-bp LTR, four 37-bp LTRs and one 40-bp LTR were detected from each of the chloroplast genomes of *Hyacinthus* (Figure 8). It was noticed that the chloroplast genome of 'City of Haarlem' did not have the 50-bp LTRs present in the chloroplast genomes of the other *Hyacinthus* cultivars. In addition, differences in LTRs were observed across chloroplast genomes from different genera. Only the chloroplast genome of *Bellevalia paradoxa* had 31-bp, 36-bp and 45-bp LTRs. The chloroplast genome of *Scilla siberica* contained the four 37-bp LTRs, one 40-bp LTR and one 50-bp LTR, but not the 32-bp LTRs present in the chloroplast genomes of *Hyacinthus*.

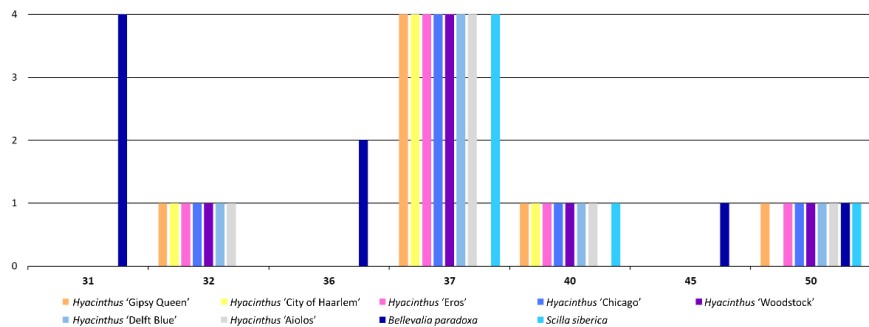

**Figure 8.** Frequency of LTRs of different lengths among the nine studied chloroplast genomes.

### 3.3. Boundary Variation

The boundaries among the four partitions in the chloroplast genome are relatively conserved among the chloroplast genomes of the seven *Hyacinthus* cultivars. At the junction JLA of all *Hyacinthus* chloroplast genomes, the gene *rps19* located in IR$_A$ with a size of 279 bp leave a 52 bp to 53 bp gap from the junction (Figure S4). Meanwhile, the gene *psbA* located in LSC was observed to be consistent in all *Hyacinthus* chloroplast genomes, with a size of 1062 bp and at a 91 bp distance from the junction. In addition, at the junction JSA, the functional gene *ycf1* was consistent in size among the chloroplast genomes of all *Hyacinthus* cultivars. The gene *ycf1* had a large portion of 4476 bp in SSC and a small portion of 984 bp in IR$_A$.

Variations were found in the junction JSB and JLB. At the junction JSB, the gene *ndhF* was consistent at 2223 bp in size. In the chloroplast genomes of *Hyacinthus* cultivars except 'Chicago', the gene *ndhF* shifted 1 bp in IR$_B$. In contrast, this gene was entirely located in SSC, leaving no distance from the junction JSB in the chloroplast genome of 'Chicago'. The pseudogene *ycf1* in IR$_B$ varied across the chloroplast genomes of the *Hyacinthus* cultivars. In the chloroplast genomes of 'Gipsy Queen', 'City of Haarlem', 'Woodstock' and 'Delft Blue', the pseudogene *ycf1* was the size of 987 bp, including a small portion of 3 bp in SSC and a

large portion of 984 bp in IR$_B$. In the chloroplast genome of 'Eros', the small portion in SSC contracted into 2 bp, giving the size of pseudogene *ycf1* as 986 bp. The shortest pseudogene *ycf1* was observed in the chloroplast genome of 'Aiolos', which the large portion in IR$_B$ contracted into 939 bp, giving the size of this pseudogene as 942 bp. Contrastingly, the longest pseudogene gene *ycf1* among the chloroplast genomes of *Hyacinthus* cultivars was found in the chloroplast genome of 'Chicago', while the small portion in SSC was extended to 23 bp, giving the size of this pseudogene as 1007 bp.

At the junction JLB, the gene *rps19* was also 279 bp in size, leaving the junction by 52 bp to 53 bp in all *Hyacinthus* chloroplast genomes. However, differences were observed from the gene *rpl22* located in LSC. This gene was 387 bp in size and just leaving from the junction by a 1-bp distance across the chloroplast genomes of the studied *Hyacinthus* cultivars, except for the one of 'Eros'. In the chloroplast genomes of 'Eros', *rpl22* was contracted by 57 bp, giving the size of the gene as 330 bp. At the same time, the distances from the junction for this gene was increased to 43 bp in this chloroplast genome.

The boundary variation among the chloroplast genomes of *Hyacinthus* cultivars showed no correlation with the flower colour. The chloroplast genomes of cultivars with different flower colours could exhibit similar patterns, such as the case of functional *ycf1* in JSA. At the same time, the chloroplast genomes of cultivars having the same flower colour could exhibit different boundary patterns, which was reflected in the case of *ndhF* and pseudogene *ycf1* in JSB.

By comparing the boundary patterns with the chloroplast genomes of *Bellevalia paradoxa* and *Scilla siberica*, the generic differentiations were observed. The size and arrangement of the genes at the four junctions were distinctively different from the chloroplast genomes of *Hyacinthus* cultivars.

*3.4. Divergence Hotspots*

Divergence hotspots were not detected from the chloroplast genomes of the *Hyacinthus* cultivars. The highest and the second highest Pi values were 0.00857 and 0.00825, which were in the region of *rps14-psaB* and *trnT-GGU-psbD*, respectively (Figure 9C). A very high percentage of identity in these two regions was found, as shown in Figure S5 that nearly no "cleavage" existed in the alignment. Therefore, these two regions could not be a suitable candidate of DNA markers for the authentication of the *Hyacinthus* cultivars.

Taking the chloroplast genomes of *Bellevalia paradoxa* and *Scilla siberica* into the calculation of Pi, the threshold value was significantly increased to 0.017 (Figure 9B). The potential regions to differentiate the three Scilloideae genera—*Hyacinthus*, *Bellevalia* and *Scilla*—were *rps16-trnQ-UUG* and *petA* in LSC, with the Pi values of 0.02125 and 0.01782, respectively.

Taking all NCBI-available chloroplast genomes of Scilloideae into the calculation of Pi, the threshold value was further increased to 0.040 (Figure 9A). Potential markers differentiating the seven Scilloideae genera—*Hyacinthus*, *Hyacinthoides*, *Barnardia*, *Scilla*, *Bellevalia*, *Albuca* and *Oziroe*—were *trnA-UGC* and *ndhA* in SSC with Pi values of 0.04232 and 0.04133, respectively.

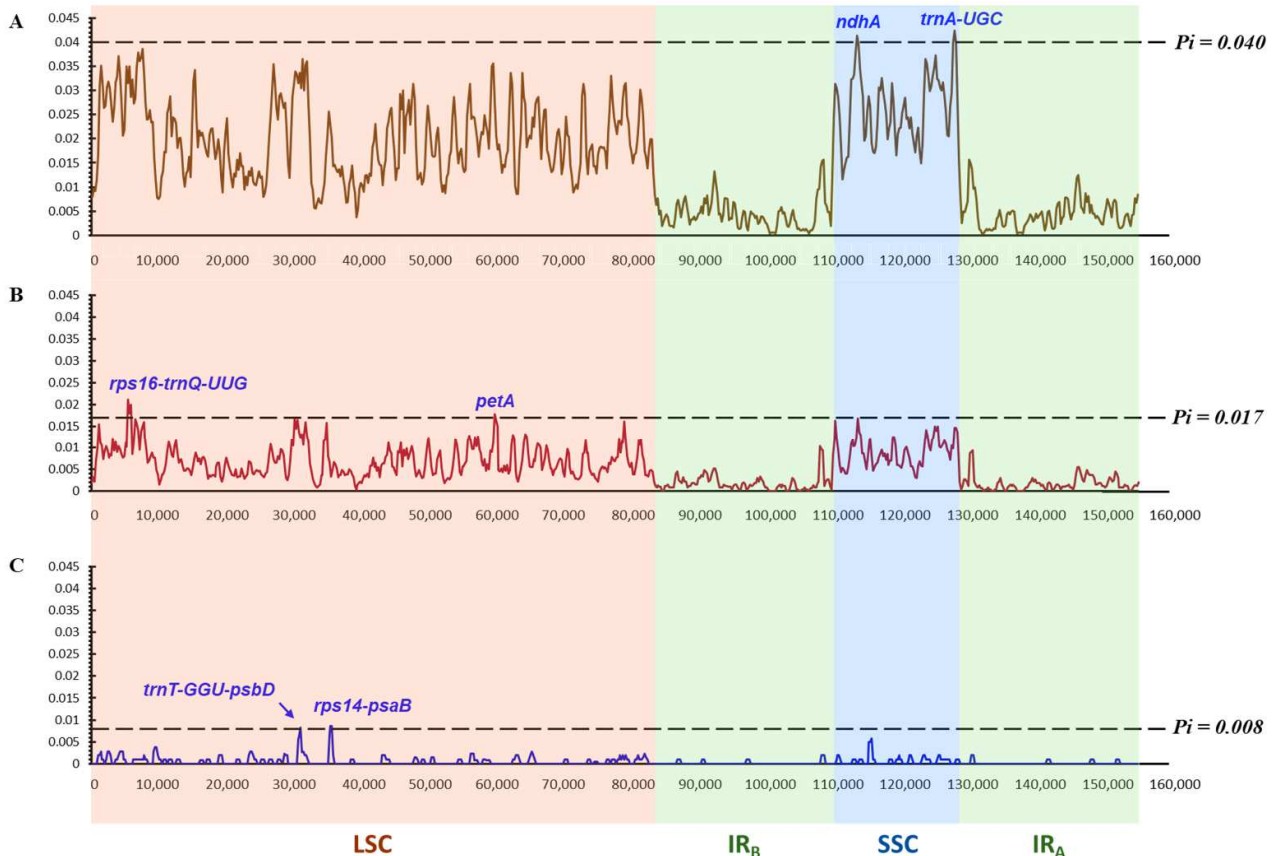

**Figure 9.** Nucleotide diversity values (Pi) of different combinations of Scilloideae chloroplast genomes. (**A**) Fifteen chloroplast genomes of Scilloideae including chloroplast genomes of *Scilla siberica*, *Bellevalia paradoxa*, *Hyacinthoides non-scripta*, *Albuca kirkii*, *Oziroe biflora*, seven *Hyacinthus* cultivars and three *Barnardia japonica*. (**B**) The nine newly de novo assembled chloroplast genomes in this study, including the chloroplast genomes of *Scilla siberica*, *Bellevalia paradoxa* and seven *Hyacinthus* cultivars. (**C**) Only the chloroplast genomes of seven *Hyacinthus* cultivars. The partitions of chloroplast genome were abbreviated as the following: Large Single Copy—LSC; IR$_B$—Inverted Repeat B; Small Single Copy—SSC; IR$_A$—Inverted Repeat A.

### 3.5. Phylogenetic Analysis

Close phylogenetic relationships among the *Hyacinthus* cultivars were observed from the comparatively short phylogenetic distances and high bootstrap values. The *Hyacinthus* cultivars were divided into two major groups, namely the group consisting of 'Aiolos', 'Delft Blue', 'Woodstock' and 'Chicago', and the group consisting of 'Eros', 'Gipsy Queen' and 'City of Haarlem'. This pattern was observed from both the ML trees calculated on the alignment of the complete chloroplast genomes (Figure 10A) and the nine extracted loci—*trnK-UUU-trnQ-UUG*, *trnS-GCU-trnG-UCC*, *petN-psbM*, *psbC*, *psbE-petL*, *ndhA*, *trnN-GUU-trnR-ACG*, *rrn23* and *trnA-UGC*—with a Pi value over 0.035 (Figure 10B). However, the positions of the cultivars within the groups were not exactly the same. In Figure 10A, 'Gipsy Queen' was sister to 'City of Haarlem', and 'Eros' was sister to these two cultivars, with bootstrap values of 100, while in Figure 10B, 'Eros', 'Gipsy Queen' and 'City of Haarlem' were clustered together instead, with bootstrap value of 99.

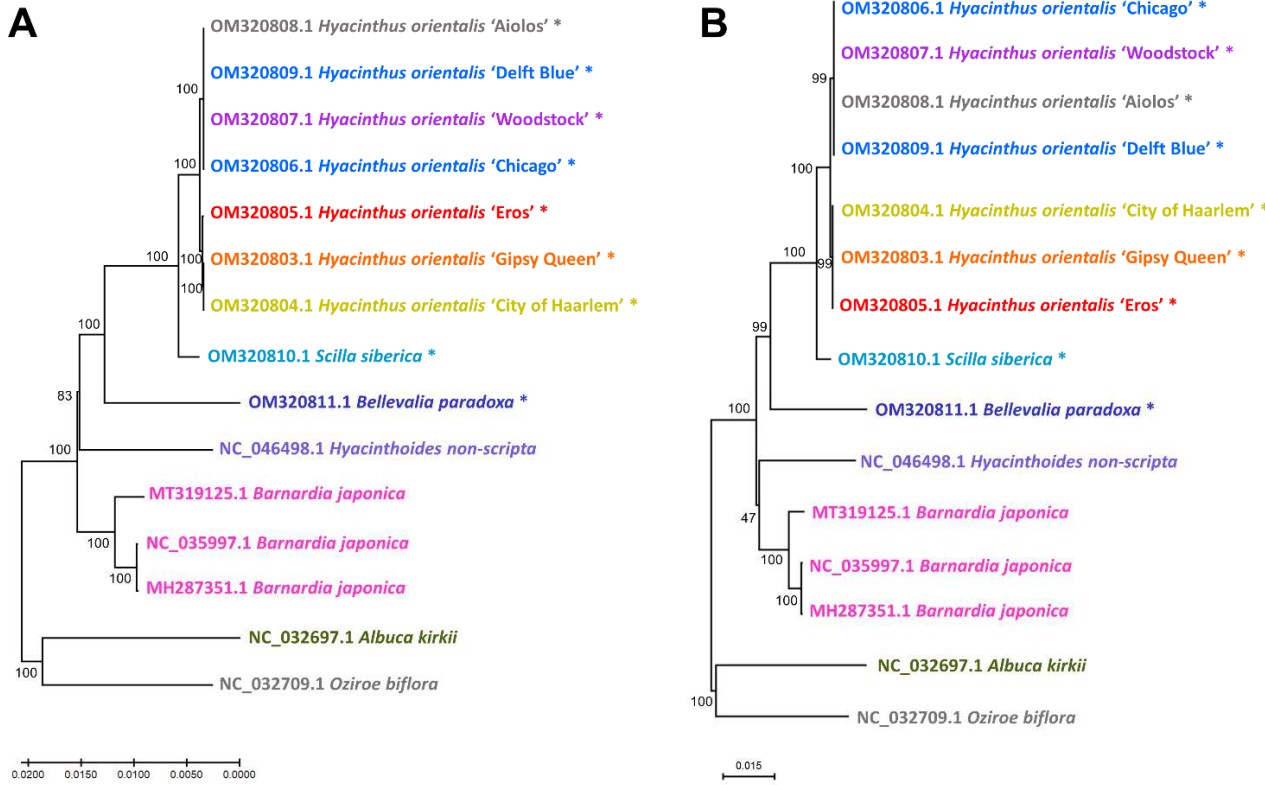

**Figure 10.** Maximum Likelihood (ML) trees based on the fifteen Scilloideae chloroplast genomes. (**A**) ML tree constructed based on the full alignment of complete chloroplast genomes. One unit in scale bar = 0.005. (**B**) ML tree constructed based on the extracted loci with Pi over 0.035 (*trnK-UUU-trnQ-UUG*, *trnS-GCU-trnG-UCC*, *petN-psbM*, *psbC*, *psbE-petL*, *ndhA*, *trnN-GUU-trnR-ACG*, *rrn23* and *trnA-UGC*). The numbers next to branch nodes are the bootstrap values. Asterisked accessions were achieved in this study. One unit in scale bar = 0.015. The accession numbers and species names were coloured according to flower colours (white is presented as grey instead).

The closest genus to *Hyacinthus* was *Scilla,* compared to *Bellevalia* from the ML tree. The genus *Hyacinthoides* was closer with *Barnardia*, but not *Hyacinthus*. It was contradictory to the treatment once *Hyacinthoides non-scripta* was classified under *Hyacinthus* [23] and fortified the generic position of *Hyacinthoides*.

The basal group of the phylogenetic tree consisted of *Albuca kirkii* and *Oziroe biflora*, which were sister to one another. The group was sister to the rest of the Scilloideae accessions.

### 3.6. Haplotype Grouping and Identification of Molecular Diagnostic Characters

Five haplotypes were identified, and the haplotype diversity value is equal to 0.8571. In the 460 variable sites, there were 130 SNPs and 330 sites of Indels (Table 4). In total, 54 Indels were found, with 6 single-nucleotide insertions, 30 single-nucleotide deletions, 1 insertion more than 1 bp and 17 deletions more than 1 bp. The longest deletion of 161 bp was found in the SSC region from 112,295 bp to 112,455 bp. Of the 130 SNPs, 75 SNPs were located in the intergenic spacers, while 55 SNPs were located within genes.

**Table 4.** Variable sites of the seven *Hyacinthus* cultivars.

| Haplotype | Locus | *psbA-trnK-UUU* | | *trnK-UUU* | | | | | *matK-rps16* | | | | | | *rps16* | *trnQ-UUG-psbK* | *psbK-psbI* | *trnS-GCU-trnG-UCC* | | ***trnG-UCC*** | *trnG-UCC-trnR-UCU* | *trnR-UCU* | *atpA* | | *atpF* | *atpF-atpH* | | *atpI-rps2* | *rps2-rpoC2* | ***rpoC2*** | | *rpoC1* | | *rpoB* | | | | |
|---|---|---|---|---|---|---|---|---|---|---|---|---|---|---|---|---|---|---|---|---|---|---|---|---|---|---|---|---|---|---|---|---|---|---|---|---|---|---|
| | Position | 1245 | 1282 | 1795 | 2458 | 2664 | 2742 | 3890 | 4395 | 4562 | 4596 | 4689 | 4690 | 4691 | 4939 | 6515 | 7067 | 7739 | 8158 | **8907-8908** | 9531-9533 | 9651 | 10,072 | 10,891 | 11,704 | 12,882 | 12,902 | 14,720 | 15,659 | **16,366** | 17,343 | 19,119 | 19,193 | 21,729 | 23,421 | 23,531 | 23,694 | 24,081 | 25,322 |
| 1 | GQ | c | c | g | c | t | g | a | t | c | a | - | - | - | a | c | g | g | a | -t | gga | t | t | a | a | t | g | - | - | a | t | t | g | t | c | c | g | c | a |
| 2 | CH | c | c | g | c | t | g | a | t | c | a | - | - | - | a | c | g | g | a | -t | gga | t | t | a | a | t | g | - | - | a | t | t | g | t | c | c | g | c | a |
| 3 | ER | c | c | g | c | t | g | a | t | c | a | - | - | - | a | c | g | g | a | **tt** | gga | t | t | a | a | t | g | - | - | g | t | t | g | t | c | c | g | c | a |
| 4 | CO | t | t | a | t | c | a | c | g | t | c | t | t | a | g | t | a | t | g | -- | tcc | c | c | g | g | - | a | a | t | g | c | c | c | a | c | t | t | t | g |
| 5 | WS | t | t | a | t | c | a | c | g | t | c | t | t | a | g | t | a | t | g | -- | tcc | c | c | g | g | - | a | a | t | g | c | c | c | a | c | t | t | t | g |
| | DB | t | t | a | t | c | a | c | g | t | c | t | t | a | g | t | a | t | g | -- | tcc | c | c | g | g | - | a | a | t | g | c | c | c | a | c | t | t | t | g |
| | AO | t | t | a | t | c | a | c | g | t | c | t | t | a | g | t | a | t | g | -- | tcc | c | c | g | g | - | a | a | t | g | c | c | c | a | c | t | t | t | g |

| Haplotype | Locus | *rpoB-trnC-GCA* | | *trnC-GCA-petN* | | ***petN-psbM*** | | | *psbM* | ***trnD-GUC-trnY-GUA*** | | ***trnY-GUA*** | *trnY-GUA-trnE-UUC* | *trnE-UUC-trnT-GGU* | *trnT-GGU-psbD* | *psbC-trnS-UGA* | | *trnS-UGA-psbZ* | | *psbZ-trnG-GCC* | *psaB* | *pafl* | | ***trnF-GAA-ndhJ*** |
|---|---|---|---|---|---|---|---|---|---|---|---|---|---|---|---|---|---|---|---|---|---|---|---|---|
| | Position | 26,541 | 26,625 | 27,382-27,383 | 27,792 | 28,087 | **28,313** | 28,637 | 28,746 | **29,334-29,339** | 30,065 | **30,688-30,694** | 30,899-30,908 | **30,982** | 31,081 | 31,168 | 31,593 | 31,749 | 35,249 | 35,298-35,304 | 35,516 | 35,596 | 36,142-36,145 | 38,441 | 43,049 | 43,124 | 43,829 | **47,709** |
| 1 | GQ | g | - | -- | c | a | t | c | t | ------ | a | **attgtta** | aattggttat | **t** | c | t | c | t | - | agtggga | g | c | atat | c | t | a | g | c |
| 2 | CH | g | - | -- | c | a | t | c | t | **aaataa** | a | **attgtta** | aattggttat | **t** | c | t | c | t | - | agtggga | g | c | atat | c | t | a | g | **t** |
| 3 | ER | g | - | -- | c | a | **a** | c | t | ------ | a | **taacaat** | aattggttat | **c** | c | t | c | t | - | agtggga | g | c | atat | c | t | a | g | c |
| 4 | CO | a | t | ag | t | - | t | t | c | ------ | - | **taacaat** | ---------- | **c** | t | c | a | c | t | tcccact | a | g | ---- | t | c | a | a | c |
| 5 | WS | a | t | ag | t | - | t | t | c | ------ | - | **taacaat** | ---------- | **c** | t | c | a | c | t | tcccact | a | g | ---- | t | c | g | a | c |
| | DB | a | t | ag | t | - | t | t | c | ------ | - | **taacaat** | ---------- | **c** | t | c | a | c | t | tcccact | a | g | ---- | t | c | g | a | c |
| | AO | a | t | ag | t | - | t | t | c | ------ | - | **taacaat** | ---------- | **c** | t | c | a | c | t | tcccact | a | g | ---- | t | c | g | a | c |

**Table 4.** *Cont.*

Gene loci (top block): *trnF-GAA-ndhJ* (47,711–47,747; 47,788–47,789) · *ndhK* (49,041) · *ndhC-trnV-UAC* (50,274–50,366) · *atpB-rbcL* (54,305–54,337) · *rbcL-accD* (56,269–56,787) · *accD* (57,943) · *accD-psaI* (58,550) · *pafII-cemA* (60,521) · *petA* (62,324–62,723) · *petA-psbJ* (62,868–62,869) · *psbE-petL* (64,219–64,223)

| Haplotype | Locus | 47,711-47,747 | 47,788-47,789 | 47,962 | 49,041 | 50,274 | 50,292 | 50,366 | 54,305-54,306 | 54,337 | 56,269 | 56,271 | 56,273 | 56,703-56,708 | 56,787 | 57,943 | 58,550 | 60,521 | 62,324 | 62,723 | 62,868-62,869 | 62,939 | 64,219-64,223 |
|---|---|---|---|---|---|---|---|---|---|---|---|---|---|---|---|---|---|---|---|---|---|---|---|
| 1 | GQ | tttctggtttttttcatactgtgctttctctcactcaa | - - | g | g | t | a | g | - - | c | c | a | a | tattta | t | t | a | - | t | c | t c | t | tttgt |
| 2 | CH | ------------------------------------ | - - | g | g | t | a | g | - - | c | c | a | a | tattta | t | t | a | - | t | c | t c | t | tttgt |
| 3 | ER | tttctggtttttttcatactgtgctttctctcactcaa | - - | g | g | - | t | g | - - | c | t | t | g | tattta | c | t | c | t | t | c | g a | t | tttgt |
| 4 | CO | tttctggtttttttcatactgtgctttctctcactcaa | a t | a | a | - | a | t | t a | a | t | t | g | ------ | c | c | c | t | c | - | t c | c | ----- |
| 5 | WS | tttctggtttttttcatactgtgctttctctcactcaa | a t | a | a | - | a | t | t a | a | t | t | g | ------ | c | c | c | c | c | c | t c | c | ----- |
| 5 | DB | tttctggtttttttcatactgtgctttctctcactcaa | a t | a | a | - | a | t | t a | a | t | t | g | ------ | c | c | c | t | c | - | t c | c | ----- |
| 5 | AO | tttctggtttttttcatactgtgctttctctcactcaa | a t | a | a | - | a | t | t a | a | t | t | g | ------ | c | c | c | t | c | - | t c | c | ----- |

Gene loci (bottom block): *psbE-petL* (64734–66897) · *rpl33-rps18* (67194–67199) · *rps18-rpl20* (67809) · *rpl20-rps12* (68912) · *clpP1* (69694–70147) · *clpP1-psbB* (71437) · *psbB-psbT* (73343–73344) · *petB* (74314) · *petD* (76275) · *petD-rpoA* (77090) · *rpoA* (77746–78243) · *rps11* (78556) · *rpl36* (78967) · *rps8-rpl14* (79305–79306) · *rpl14-rpl16* (80483) · *rpl16* (80905–81847) · *rpl22-rps19* (83198–83212)

| Haplotype | Locus | 64734 | 64962 | 65097 | 66897 | 67194-67199 | 67809 | 68912 | 69694 | 69883 | 69975 | 70147 | 71437 | 73343-73344 | 73373 | 74314 | 76275 | 77090 | 77746 | 78243 | 78556 | 78967 | 79305-79306 | 79930 | 80483 | 80905 | 81004-81008 | 81468 | 81662 | 81738 | 81847 | 83198-83212 |
|---|---|---|---|---|---|---|---|---|---|---|---|---|---|---|---|---|---|---|---|---|---|---|---|---|---|---|---|---|---|---|---|---|---|
| 1 | GQ | g | c | g | - | atgtgt | a | - | a | t | t | - | - | - t | a | a | t | g | g | a | c | g | ca | t | g | g | ----- | c | t | - | c | gatttctttatcata |
| 2 | CH | g | c | g | - | atgtgt | a | - | a | t | t | - | - | - t | a | a | t | g | g | a | c | g | ca | t | g | g | ----- | c | t | - | c | gatttctttatcata |
| 3 | ER | g | c | g | - | atgtgt | a | - | a | t | t | t | t | tt | a | g | t | g | g | a | c | g | - | g | g | g | ----- | c | c | - | c | --------------- |
| 4 | CO | a | a | t | t | ------ | - | t | - | - | c | - | - | -- | t | a | g | a | a | g | t | a | -- | t | a | a | ttaat | t | t | a | t | gatttctttatcata |
| 5 | WS | a | a | t | t | ------ | - | t | - | - | c | - | - | -- | t | a | g | a | a | g | t | a | -- | t | a | a | ttaat | t | t | a | t | gatttctttatcata |
| 5 | DB | a | a | t | t | ------ | - | t | - | - | c | - | - | -- | t | a | g | a | a | g | t | a | -- | t | a | a | ttaat | t | t | a | t | gatttctttatcata |
| 5 | AO | a | a | t | t | ------ | - | t | - | - | c | - | - | -- | t | a | g | a | a | g | t | a | -- | t | a | a | ttaat | t | t | a | t | gatttctttatcata |

**Table 4.** *Cont.*

| Haplotype | | *rps19* | *ycf2* | | *ndhB-rps7* | *trnR-ACG-trnN-GUU* | | | | *ycf1* | *ndhF* | | *ndhF-rpl32* |
|---|---|---|---|---|---|---|---|---|---|---|---|---|---|
| | Position | 83248 | 86579 | 90294 | 96775 | 107482-107490 | 107915 | 107924 | 107931 | **109716** | 110054 | 110285 | 112295-112398 |
| 1 | GQ | t | t | t | t | --------- | - | t | g | - | c | a | taaaaaagtatagtaagtaaaaacaattatatcaaaacagtagaatatggatcataatatgcattgataaatcgactaaaaaaaaaagacctaattattctaat |
| 2 | CH | t | t | t | t | --------- | - | t | g | - | c | a | taaaaaagtatagtaagtaaaaacaattatatcaaaacagtagaatatggatcataatatgcattgataaatcgactaaaaaaaaaagacctaattattctaat |
| 3 | ER | t | t | t | t | --------- | - | t | g | - | c | a | ------------------------------------------------------------------------------------------------------- |
| 4 | CO | - | c | c | c | agaaaaaag | c | c | t | **t** | t | g | taaaaaagtatagtaagtaaaaacaattatatcaaaacagtagaatatggatcataatatgcattgataaatcgactaaaaaaaaaagacctaattattctaat |
| | WS | - | c | c | c | agaaaaaag | c | c | t | - | t | g | taaaaaagtatagtaagtaaaaacaattatatcaaaacagtagaatatggatcataatatgcattgataaatcgactaaaaaaaaaagacctaattattctaat |
| 5 | DB | - | c | c | c | agaaaaaag | c | c | t | - | t | g | taaaaaagtatagtaagtaaaaacaattatatcaaaacagtagaatatggatcataatatgcattgataaatcgactaaaaaaaaaagacctaattattctaat |
| | AO | - | c | c | c | agaaaaaag | c | c | t | - | t | g | taaaaaagtatagtaagtaaaaacaattatatcaaaacagtagaatatggatcataatatgcattgataaatcgactaaaaaaaaaagacctaattattctaat |

| Haplotype | | *ndhF-rpl32* | *ndhF-rpl32* | *rpl32-trnL-UAG* | | *ccsA-ndhD* | | *ndhD* | *psaC-ndhE* | | *ndhE-ndhG* | *ndhI* | *ndhI-ndhA* | *ndhA* | | | | *ndhH* | *rps15* | *rps15-ycf1* | |
|---|---|---|---|---|---|---|---|---|---|---|---|---|---|---|---|---|---|---|---|---|---|
| | Position | 112399-112455 | 112621 | 113173 | 113555 | **115044-115048** | 115050 | 115345 | 117286-117290 | 117432-117438 | 117922 | 119067 | 119459 | 120327 | 120328 | **120555** | 120815 | 120971 | 122856 | 123081 | 123607 | 123669 | 123670 |
| 1 | GQ | gaattcatttgtagtaattacctaattcattgcagaatttattggattccaatccaa | c | - | a | aggaa | c | g | ----- | ------- | t | g | g | - | - | - | a | c | a | g | c | c | a |
| 2 | CH | gaattcatttgtagtaattacctaattcattgcagaatttattggattccaatccaa | c | - | a | aggaa | c | g | ----- | ------- | t | g | g | - | - | - | a | c | a | g | c | c | a |
| 3 | ER | ------------------------------------------------------- | c | - | a | ttcct | c | g | ----- | ------- | t | g | g | - | - | a | a | c | a | g | c | c | a |
| 4 | CO | gaattcatttgtagtaattacctaattcattgcagaatttattggattccaatccaa | t | t | g | ttcct | g | t | atgtg | atttata | a | a | t | a | a | - | g | t | g | t | g | - | - |
| | WS | gaattcatttgtagtaattacctaattcattgcagaatttattggattccaatccaa | t | t | g | ttcct | g | t | atgtg | atttata | a | a | t | a | a | - | g | t | g | t | g | - | - |
| 5 | DB | gaattcatttgtagtaattacctaattcattgcagaatttattggattccaatccaa | t | t | g | ttcct | g | t | atgtg | atttata | a | a | t | a | a | - | g | t | g | t | g | - | - |
| | AO | gaattcatttgtagtaattacctaattcattgcagaatttattggattccaatccaa | t | t | g | ttcct | g | t | atgtg | atttata | a | a | t | a | a | - | g | t | g | t | g | - | - |

**Table 4.** *Cont.*

| Haplotype | Locus | Position | 125016 | 125184 | 125908 | 126456 | 127892 | 129981 | 129988-129989 | 130415-130423 | 141137 | 147618 | 151333 | 154655 | |
|---|---|---|---|---|---|---|---|---|---|---|---|---|---|---|---|
| | | | *ycf1* | | | | | *trnN-GUU-trnR-ACG* | | *trnR-ACG-rrn5* | *rps7-ndhB* | *ycf2* | | *rps19-psbA* | |
| 1 | GQ | | a | c | g | c | a | c | - a | - - - - - - - - - | a | a | a | a | |
| 2 | CH | | a | c | g | c | a | c | - a | - - - - - - - - - | a | a | a | a | |
| 3 | ER | | a | c | g | c | a | c | - a | - - - - - - - - - | a | a | a | a | |
| 4 | CO | | g | t | a | a | c | a | g g | t t t t t c t c t | g | g | g | - | |
| 5 | WS | | g | t | a | a | c | a | g g | t t t t t c t c t | g | g | g | - | |
| | DB | | g | t | a | a | c | a | g g | t t t t t c t c t | g | g | g | - | |
| | AO | | g | t | a | a | c | a | g g | t t t t t c t c t | g | g | g | - | |

The cultivar epithets were abbreviated as the following: 'Gipsy Queen'—GQ; 'City of Haarlem'—CH; 'Eros'—ER; 'Chicago'—CO; 'Woodstock'—WS; DB—'Delft Blue'; 'Aiolos'—AO. Cells containing MDCs were coloured in blue, pink and yellow, which represent the MDCs of haplotypes 4, 3 and 2, respectively. Cells in orange and purple represent the variable sites separating the haplotype groups 1 + 2 and 3 + 4 + 5, respectively. The alignment positions and loci of these MCDs and variable sites were bolded. Deletion of a nucleotide was presented by the symbol "-". Grey cells contain no data.

The median-joining network of the seven *Hyacinthus* cultivars is shown in Figure 11. The seven cultivars were grouped in five haplotypes. While haplotype 5 consisted of 'Delft Blue', 'Woodstock' and 'Aiolos', other haplotypes were represented by one cultivar only. Haplotype 4 ('Chicago') had the least difference from haplotype 5 as only one variable site—the insertion in SSC—was detected. Haplotype 1 ('Gipsy Queen') and 2 ('City of Haarlem') were close to one another, with only a few variations.

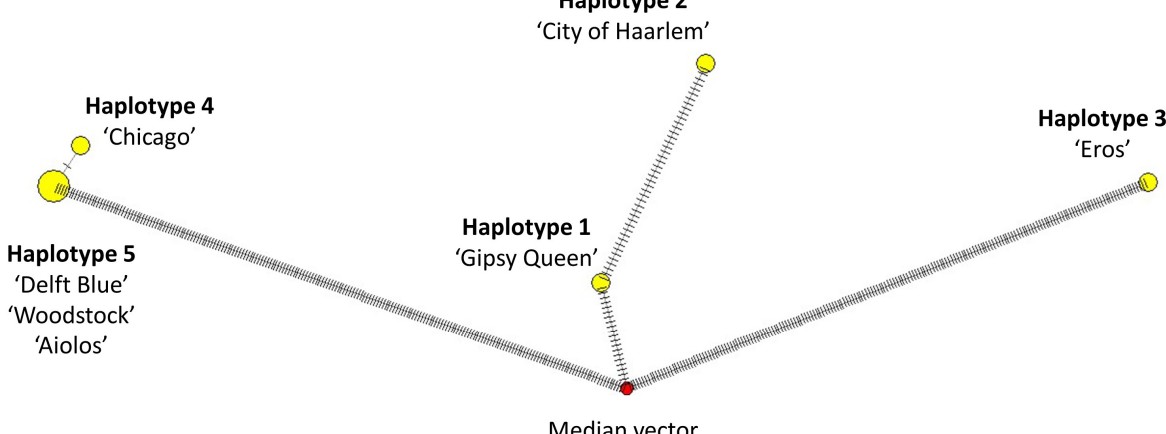

**Figure 11.** Median-joining network of the seven *Hyacinthus* cultivars. Median vector represents the sequences of cultivars, which are extinct ancestors or not sampled in this study. One hedgemark represents one variable site between the linked objects.

All variable sites can differentiate the five haplotypes into two groups, as haplotypes 1 + 2 + 3 and 4 + 5, except for the MDCs highlighted in colours (Table 4). The insertion of thymine (t) at alignment position 109,716 bp in *ycf1* was the only MDC for haplotype 4 ('Chicago') highlighted in blue. The MDCs for haplotype 2 ('City of Haarlem') highlighted in yellow included 44 variable sites, namely the insertion of sequence "aaataa" in *petN-psbM* (29,334 bp to 29,339 bp), the base substitution of thymine instead of cytosine (c) at 47,709 bp in *trnF-GAA-ndhJ* and also a 37-bp deletion in *trnF-GAA-ndhJ* (47,711 to 47,747 bp). The MDCs for haplotype 3 ('Eros') highlighted in pink consisted of 188 variable sites, including the insertions of thymine at 8907 bp, 70,147 bp, 71,437 bp and 73,343 bp; the insertion of adenine (a) at 120,555 bp, six single-base substitutions at 28,313 bp, 50,292 bp, 62,868 bp, 62,869 bp, 74,314 bp and 81,662 bp; and the deletions at 79,930 bp, 83,198 bp to 83,212 bp (15-bp) and 112,295 bp to 112,455 bp (161-bp). There were 21-bp variable sites differentiating haplotypes 1 + 2 (in orange) and 3 + 4 + 5 (in purple), which were distinct from other non-highlighted variable sites.

## 4. Discussion

### 4.1. Chloroplast Genomes Serve as New Evidence in the Phylogeny of Hyacinthus cultivars

The study of Hu et al. in 2015 [9] investigated phylogenetic relationships among 29 *Hyacinthus* cultivars using twelve ISSR molecular markers, revealing some correlation with flower colours. Genetic similarity and genetic distance were calculated based on the ISSR banding patterns and used for the construction of the UPGMA tree. Cultivars with flowers of the same colour were almost clustered together [9].

Although our study cannot prove the correlation between flower colour and phylogenetic relationship due to the limited sample size, informative molecular resources were generated. Five cultivars—'Gipsy Queen', 'City of Haarlem', 'Aiolos', 'Delft Blue' and 'Woodstock'—were studied in both our study and Hu's study in 2015 [9]. The orange-flower cultivar 'Gipsy Queen' was closely related to the yellow-flower cultivar 'City of Haarlem' in our ML tree. In contrast, these two cultivars were not clustered together in Hu's UPGMA tree. 'Gipsy Queen' was sister to the group of white-flower cultivars ('Aiolos', 'White Pearl' and 'Carnegie'), while 'City of Haarlem' was sister to the group consisting of the yellow-



flower cultivar 'Gipsy Princess' and the red-flower cultivar 'Red magic' [9]. Interestingly, in our study, 'Aiolos' was grouped with the blue-flower ('Chicago' and 'Delft Blue') and purple-flower ('Woodstock') cultivars, but not 'Gipsy Queen'. Our results align with the record of the *International Checklist for Hyacinths and Miscellaneous Bulbs*, that 'Delft Blue' was indicated as the female parent of 'Aiolos' [19].

### 4.2. Timeline of Colour Evolution and Phylogenetic Relationship of Hyacinthus Cultivars

The original flower colour of the hyacinth is blue [16,73,74]. The wild population of hyacinths has blue blossoms [5,16], while the subspecies *albulus* (Jord.) Nyman naturalised in France has white flowers [5,19,21]. The enrichment of flower colours began with the introduction for cultivation. According to Kersten [16] and Darlington [20], new flower colours of the hyacinth cultivar were recorded as of the late sixteenth century. In 1582, a white cultivar was raised from seed [16]. In 1596, double-flower cultivars in purple, blue and white were recorded [16]. In 1614, single-flower cultivars in red were recorded [16]. The latest flower colour to be bred was yellow near the end of the eighteenth century, with the first record in a catalogue in 1767 [16].

Being an ICRA as appointed by the ISHS Commission for Nomenclature and Cultivar Registration, KAVB is responsible for the registration of *Hyacinthus* cultivars [38]. In the database of KAVB [39] and its latest publication *International Checklist for Hyacinths and Miscellaneous Bulbs* in 1991 [19], the registration date and registrants for each hyacinth cultivar were recorded. However, the date of cultivar registration provides no indication of the evolution of the floral pigments, as the cultivar may have existed long before the registration date. From the ML tree, the blue cultivars 'Delft Blue' and 'Chicago' clustered with the purple cultivar 'Woodstock' and the white cultivar 'Aiolos' (Figure 10). However, the dates of registration for these two blue cultivars have a gap of 66 years (Table 5) [19]. This suggests that a temporal distance of cultivar registration may not be related to the phylogenetic distances between cultivars. New cultivars could be bred from older cultivars and hence demonstrate shorter phylogenetic distances.

**Table 5.** Information of the registered *Hyacinthus* cultivars from KAVB.

| Cultivar | Date of Registration | Registrant | Description | Perianth | Forcing Time | Reference |
|---|---|---|---|---|---|---|
| 'City of Haarlem' | 1893/12/31 | J.H. Kersten | Soft primrose-yellow | Single | Late forcing | [19,39] |
| 'Gipsy Queen' | 1927/dd/mm * | G. van der Meij | Dark salmon (421/1) and apricot (608/1) | Single | Mid early forcing | [19,39] |
| 'Eros' | 1935/12/31 | M. Veldhuyzen-van Zanten | Deep pink | Double | Early forcing | [19,39] |
| 'Delft Blue' | 1944/12/31 | J.W.A. Lefeber | Soft lilac-blue | Single | Early forcing | [19,39] |
| 'Aiolos' | 1985/12/31 | I.V.T. | Ivory white, outside tube creamy white, grey-yellow blotch at top | Single | / | [19,39] |
| 'Woodstock' | 1992/9/17 | Jac. Prins et Zn. B.V. | / | Single | / | [39] |
| 'Chicago' | 2009/6/17 | Fa. M.C. Zonneveld et Zn. | / | Single | / | [39] |

* No exact date was recorded.

The other three cultivars, i.e., 'Eros', 'Gipsy Queen' and 'City of Haarlem' were clustered into a group (Figure 10). In our samples, these three registered cultivars are the oldest three, while the yellow cultivar 'City of Haarlem' is the oldest registered cultivar, registered in 1893 [19]. The second oldest one is the orange cultivar 'Gipsy Queen' registered in 1927, followed by the double, deep-pink cultivar 'Eros' registered in 1935 [19]. The close phylogenetic relationship among these three cultivars was believed to be related to the evolution of floral pigments, not the registration date.

### 4.3. Differences among the Chloroplast Genomes of Hyacinthus Cultivars

The seven newly assembled chloroplast genomes of the *Hyacinthus* cultivars were highly conserved, regardless of their sizes, structures and gene contents. All these seven chloroplast genomes shared identical gene number and gene content, as shown in Tables 2 and 3. It is not surprising because all the cultivars were derived from the same species, *Hyacinthus orientalis* L. Despite long-term artificial selection for their appearance through domestication, their chloro-

plast genomes were not altered drastically since the chloroplast genome do not undergo recombination [75]. This also applies to the chloroplast genomes of the three *Camellia oleifera* cultivars 'Huashuo', 'Huaxin' and 'Huajin', where 133 genes including 88 mRNA genes, 37 tRNA genes and 8 rRNA genes were identically annotated in each of their chloroplast genomes [76]. The size variation of these *Camellia* chloroplast genomes was also insignificant, as there was only a 10-bp difference [76]. In the case of the *Hyacinthus* cultivars, there was a 183-bp difference in chloroplast genome size, which is higher than the case of *Camellia oleifera*, but not significant.

The expansion and contraction of IR regions from evolutionary events were anticipated as the major driver of the variation in genome size [77–80]. In our study, significant differences were only observed in the junctions JLB and JSB among the chloroplast genomes of the *Hyacinthus* cultivars (Figure S4). The smallest chloroplast genome of a hyacinth cultivar was the one of 'Eros'; the decrease in genome size was noted to be related to the deletions in SSC (Tables 2 and 4), but not IR. The variation of chloroplast genome size was also reportedly not due to the expansion or contraction of IRs [81,82].

From the sliding window analysis, the nucleotide diversity threshold among the *Hyacinthus* cultivars was too low (Pi = 0.008). The two weak hotspots—*trnT-GGU-psbD* (Pi = 0.00825) and *rps14-psaB* (Pi = 0.00857)—even showed a high percentage identity in the alignment visualised by mVISTA (Figure S5), barely had any differentiating power and were not expected to be used as a barcode marker to differentiate *Hyacinthus* cultivars. Contrastingly, twelve hotspots were identified with a nucleotide diversity threshold at 0.2 among the chloroplast genome of three *Utricularia amethystina* morphotypes, which blossom in purple, yellow and white [83].

Despite the high similarity, differences were discovered from the results of sequence repeat analysis. The chloroplast genome of 'Eros' had a significantly high number of AT/AT repeats at 10, which is the highest among all chloroplast genomes assembled in this study. Meanwhile, the AAGC/CTTG repeat and ACTAT/AGTAT were identified only in the chloroplast genome of 'Eros'. These could become potential molecular markers for this cultivar. Similarly, the absence of 50-bp LTRs was observed in the chloroplast genome of the cultivar 'City of Haarlem', which is a distinctive molecular character from the chloroplast genome of the other six *Hyacinthus* cultivars.

To increase the resolution on the differences, a haplotype grouping was performed. A total of 460 variable sites were identified, differentiating the *Hyacinthus* cultivars into five haplotypes. The cultivars 'Delft Blue', 'Woodstock' and 'Aiolos' were grouped as one haplotype, while the other four were differentiated as separate haplotypes. From the variable sites including SNPs and Indels, the MDCs were identified to distinguish these cultivars (Table 4). The identification of crop cultivars extensively utilised molecular markers including SNPs [84–88] and Indels [88–91]. Meanwhile, the MDCs could be a kind of evidence for taxonomical treatment. Jafari et al. [92] conducted a phylogenetic study of the *Bellevalia* species using the chloroplast regions *rbcL*, *matK*, *trnL* intron and *trnL-trnF* intergenic spacer, and polymorphic positions were identified from the alignment. They proposed a revised infrageneric classification of *Bellevalia* combining the evidence of morphological characters and phylogenetic results [92]. Therefore, the MDCs for the *Hyacinthus* haplotypes may contribute to the identification and phylogenetic study of *Hyacinthus* cultivars.

It was important to identify molecular markers for *Hyacinthus* cultivars, since the cultivars can hardly be differentiated by vegetative morphological characters. As shown in Figure 2, the cultivars with white-to-yellow flowers generally have beige-to-white outer tunics, while the cultivars that blossom in red, pink, blue and purple usually have dark purple outer tunics. The orange-flower cultivar 'Gipsy Queen' can be identified with its silvery purple outer tunics, but this character can also be seen in the yellow-flower cultivar 'Gipsy Princess'. In the nonflowering season, the identity of *Hyacinthus* cultivars can hardly be distinguished unless properly labelled, which would cause severe inconveniences and problems in the breeding and propagation of new cultivars.

We attempted to search for molecular markers for the studied *Hyacinthus* cultivars. Although we cannot identify a specified region with high nucleotide diversity to differentiate all these cultivars, the analysis of sequence repeats and the identification of MDCs revealed some potential molecular markers to identify specified cultivars.

*4.4. Understanding the Phylogenetic Relationship Contributes to Cultivar Breeding*

Breeding new cultivars is important in the industry of floriculture, which may bring potential economic income and better production performance. In 2005, three mother bulbs of a new "black-flowered" cultivar 'Midnight Mystique' were sold at £ 50,000 per piece [93]. Cultivars with new flower colours can be sold at a good price. Meanwhile, the hyacinth is susceptible to various kinds of diseases, for example, bacterial infection by *Xanthomonas hyacinthi*, fungal infection by *Penicillium* and viral infection by Hyacinth mosaic virus [7]. It is also affected by physiological defects such as "straw-nails" (uppermost florets abortion) [7] (see the voucher specimen of 'City of Haarlem' K. H. Wong 139 in Figure S1), "splitting" (inflorescence detachment) [7] and "stem topple" (scape collapse) [94]. Hence, there is a need to produce new cultivars to overcome or resist these diseases and defects.

Contributing to over 90% of hyacinth production worldwide, the Netherlands is the major hyacinth-producing hub [7,16,73]. From the statistics issued by the governmental statistical authority of the Netherlands, StatLine, the production of hyacinths has been gradual increasing in the past twenty years, from 66,544,000 pcs in 2002 to 84,786,000 pcs in 2021 [95], underlying the important and irreplaceable role of the hyacinth in the economy of the Netherlands.

The seven *Hyacinthus* cultivars investigated in our studied were commercialised. Among them, the orange-flower cultivar 'Gipsy Queen' is one of the most important parental plants of other *Hyacinthus* cultivars. The crossbred offspring blossom in orange ('Odysseus'), yellow ('Hektor', 'Helena', 'Hellas', 'Herakles', 'Hermes', 'Hermione' and 'Yellow Queen') and rose-red ('Morpheus', 'Orpheus', 'Prometheus', 'Proteus' and 'Theseus') [19]. The blue-flower cultivar 'Delft Blue' is also a common ancestor of numerous cultivars with a wide range of flower colours, including blue ('Blue Star', 'City of Bradford', 'Midas' and 'Sylvester'), purple ('Angelique', 'Purple Beauty', 'Minos', 'Weijers Favourite' and 'Blue Peter'), pink ('China Pink' and 'Zeus') and white ('Aiolos', 'Atlas' and 'Pallas') [19]. According to Hu et al. in 2015 [9], hybrids crossbred by parents with different flower colours was suggested according to the longer phylogenetic distances. The newly assembled chloroplast genomes in this study provide more molecular information about the Scilloideae species and will serve as genetic resources for the cultivation and breeding of *Hyacinthus* cultivars.

*4.5. Future Direction of Hyacinthus Cultivar Phylogeny*

With technological advancements, whole-genome sequencing (WGS) and assembly have become more feasible and affordable [96]. WGS has been applied in the phylogenomic study of other angiosperms, e.g., family Chrysobalanaceae [97], genus *Artocarpus* [98], genus *Rhododendron* [99] and genus *Asclepias* [100].

The study of Malé et al. in 2014 [97] utilised WGS to resolve the phylogeny of a systematically recalcitrant family Chrysobalanaceae, and the results strongly supported the paraphyly of the genus *Licania* and suggested systematic revision of this genus. Meanwhile, the study demonstrated genome skimming to find the best-fit genome with the highest phylogenetic resolution by shallow ($1\times$) shotgun sequencing [97]. This method could be applied in studying the phylogenomics of *Hyacinthus* cultivars in order to generate informative genomic data with effectiveness in both time and resource.

By considering chloroplast, mitochondrial and nuclear loci, the study of Blischak et al. in 2014 [101] applied WGS with extremely low coverage ($0.005\times$–$0.007\times$) to identify potential markers for the phylogenetically complicated genus *Penstemon*. Their approach could be adopted to identify potential molecular markers of *Hyacinthus* cultivars apart from ISSR.

Having the potential to increase the resolution and support in phylogenetic study [100], WGS could be a future direction in resolving the phylogeny of *Hyacinthus* cultivars through comparatively analysing genomic data from nuclear, chloroplast and mitochondria. Anthropocentrically beneficial plants are always in the spotlight for WGS [102]. *Hyacinth orientalis*, being a major flower crop with economic importance, a species with significant taxonomical importance, and a potential medicinal plant with promising anticancer and immunomodulatory activities [103], should also be considered for WGS.

## 5. Conclusions

This is the first study reporting the chloroplast genomes of the genus *Hyacinthus* L., *Bellevalia* Lapeyr. and *Scilla* L. where phylogenetic relationships among *Hyacinthus* cultivars were revisited using chloroplast genomes. A total of nine chloroplast genomes were de novo sequenced and assembled using Illumina sequencing technology, including the chloroplast genomes of *Bellevalia paradoxa*, *Scilla siberica* and seven *Hyacinthus orientalis* L. cultivars ('Gipsy Queen', 'City of Haarlem', 'Eros', 'Chicago', 'Woodstock', 'Delft Blue' and 'Aiolos'). The chloroplast genomes of seven *Hyacinthus* cultivars were highly conserved in terms of structure, gene order and gene contents. From the results of the sequence repeat analysis and MDCs identification, significant differences among the *Hyacinthus* cultivars were identified, which may be selected as potential molecular markers for authentication. Phylogenetic trees based on chloroplast genomes demonstrated alternative phylogenetic positions of *Hyacinthus* cultivars compared to the previous study using ISSR. The results of this study contribute to the preservation of hyacinth germplasm and the breeding programme of new *Hyacinthus* cultivars. The study also enriches the understanding of genomic information of the subfamily Scilloideae *sensu* APG IV, which may serve as useful information in the phylogenetic study of this subfamily.

**Supplementary Materials:** The following supporting information can be downloaded at: https://www.mdpi.com/article/10.3390/horticulturae8050453/s1, Figure S1: Voucher specimens of K. H. Wong 127, 139, 144, 047, 114, 118, 120, 126 and 141; Figure S2: Microscopic observation of the seven studied *Hyacinthus* cultivars; Figure S3: Circular genome maps of all studied specimens; Figure S4: Visualisation of the boundary variations across the nine newly assembled chloroplast genomes; Figure S5: Alignment visualisation of fifteen Scilloideae chloroplast genomes using *Hyacinthus orientalis* L. 'Gipsy Queen' as reference.

**Author Contributions:** Conceptualisation, K.-H.W.; methodology, K.-H.W. and B.L.-H.K.; software, K.-H.W., H.-Y.W., T.-Y.S. and B.L.-H.K.; validation, K.-H.W. and H.-Y.W.; formal analysis, K.-H.W.; investigation, K.-H.W.; resources, K.-H.W., D.T.-W.L., J.H.-L.H. and P.-C.S.; data curation, K.-H.W.; writing—original draft preparation, K.-H.W.; writing—review and editing, J.H.-L.H., P.-C.S., D.T.-W.L., T.-Y.S., H.-Y.W., B.L.-H.K. and G.W.-C.B.; visualisation, K.-H.W.; supervision, P.-C.S., J.H.-L.H. and D.T.-W.L.; project administration, P.-C.S., J.H.-L.H. and D.T.-W.L. All authors have read and agreed to the published version of the manuscript.

**Funding:** The research work was supported by a donation fund from Wu Jieh Yee Charitable Foundation Limited.

**Institutional Review Board Statement:** Not applicable.

**Informed Consent Statement:** Not applicable.

**Data Availability Statement:** The data presented in this study are openly available in the NCBI GenBank (https://www.ncbi.nlm.nih.gov (accessed on 1 March 2022)) with the accession number OM320803 to OM320811.

**Conflicts of Interest:** The authors declare no conflict of interest. The donors had no role in the design of the study; in the collection, analyses or interpretation of data; in the writing of the manuscript, or in the decision to publish the results.

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
