# Peer review of "Characterisation of the Complete Chloroplast Genomes of Seven Hyacinthus orientalis L. Cultivars: Insights into Cultivar Phylogeny"

_horticulturae, doi:10.3390/horticulturae8050453_

Round 1

Reviewer 1 Report

The reviewed manuscript is well written and provides novel data for the community. Just a few major and minor issues should be fixed before possible acceptance.

Major issues

It’s not clear, at least from text, if cultivars can be delimited using the superbarcoding approach. It’s worth checking the presence of molecular diagnostic characters for each cultivar. It can be easily done using FASTACHAR software. The presence of MDCs can be visualized on Pi diversity figures. Obtained values can be discussed with literature. 

The phylogenetic dataset is very limited, and especially in the case of plastome dataset should be expanded with sequenced plastomes available in the genbank database. The second reason to expand this part of analysis is paraphylly of Bellevalia, which shouldn’t be used as an outgroup in this case.

Moreover, it’s also unlikely the whole plastome has a homogeneous evolutionary model (which was not given in the Methods section). The alignment should be partitioned according to the optimal substitution model. Authors could use a Phylosuite software to easily perform more advanced phylogenetic analyses.

Minor comments

Figure 9 - I would move it to supplementary materials, as the only significant changes is moving ycf1 in “Aiolos” cultivar and ndhF in Bellevalia. 

Figure 10. The second IR should be removed from analysis to not duplicate variation results.

Figures  11 & 12.. Since Bellevalia is resolved as paraphyletic, the different outgroup taxa should be used.

Author Response

Major issues

Point 1: It’s not clear, at least from text, if cultivars can be delimited using the superbarcoding approach. It’s worth checking the presence of molecular diagnostic characters for each cultivar. It can be easily done using FASTACHAR software. The presence of MDCs can be visualized on Pi diversity figures. Obtained values can be discussed with literature. 

Response: Thank you for your suggestions. a full table of variable sites including SNPs and Indels are included in the manuscript as shown in Table 4, in which the Molecular Diagnostic Characters (MDCs) were highlighted.

Point 2: The phylogenetic dataset is very limited, and especially in the case of plastome dataset should be expanded with sequenced plastomes available in the genbank database. The second reason to expand this part of analysis is paraphylly of Bellevalia, which shouldn’t be used as an outgroup in this case.

Response: Thank you for your suggestions. The phylogenetic dataset is expanded to include all NCBI-available Scilloideae chloroplast genomes. Meanwhile, Scilla siberica and Bellevalia paradoxa are mentioned as “Scilloideae species”, but not “outgroup”.

Point 3: Moreover, it’s also unlikely the whole plastome has a homogeneous evolutionary model (which was not given in the Methods section). The alignment should be partitioned according to the optimal substitution model. Authors could use a Phylosuite software to easily perform more advanced phylogenetic analyses.

Response: Thank you for your suggestions. On top of the full alignment of the complete chloroplast genomes, the loci with Pi over 0.035—trnK-UUU-trnQ-UUG, trnS-GCU-trnG-UCC, petN-psbM, psbC, psbE-petL, ndhA, trnN-GUU-trnR-ACG, rrn23, trnA-UGC—were extracted and used for the construction of ML trees.

Minor comments

Point 4: Figure 9 - I would move it to supplementary materials, as the only significant changes is moving ycf1 in “Aiolos” cultivar and ndhF in Bellevalia

Response: Agreed. We further verified the position and length of ycf1 in ‘Aiolos’ at the boundary of SSC-IRA, and we found there is an annotation mistake which should not have such “significant change”. The annotation is updated in NCBI Genbank, and the figure has been revised and moved to supplementary materials.

Point 5: Figure 10. The second IR should be removed from analysis to not duplicate variation results.

Response: The first IR is IRB and the second one is IRA. The x-axis indicated the position within the chloroplast genome.

Point 6: Figures 11 & 12. Since Bellevalia is resolved as paraphyletic, the different outgroup taxa should be used.

Response: The term “outgroup” for both Bellevalia and Scilla species is removed. Instead, “Scilloideae species” is given for these two species.

Reviewer 2 Report

The work by Wong et al entitled “Characterisation on the complete chloroplast genomes of seven Hyacinthus orientalis L. cultivars: Insights into cultivar phylogeny”, provides the chloroplast genome sequences of 7 cultivars along with two out-groups. They used conventional bioinformatic analyses to address the phylogenetic relationships between the sample materials. The approach conducted is valid to characterize the molecular diversity between cultivars, especially when morphological characters are not enough.

One of my concerns is the use of the color of flowers as a character to discuss the groups found. Given that several colors are unique, and for that, autapomorphic, this precludes its use for cladistic analysis. For grouping, you need by definition state characters that are shared by two or more terminals. Subheading 4.1 is conceptually incorrect. Also, the discussion about this point should be reconsidered. Moreover, is there any paper where the variation in flower color is consistent with cpDNA (at the infrageneric scale)?

Given the phylogenetic proximity of samples, and probably the domestication process underlying these cultivars, the diversity found was very slow. This is somehow expected for me, using cpDNA. The authors took a lot of effort into describing the lack of difference in genes and regions size, which for me is not promising from the beginning. Are other infrageneric studies within flowering plants where the size (insertions and deletions) are meaningful?. For me, SNPs are always more promising. Although the phylogenetic tree is partially informative, there is a lack of resolution is in the Chicago and relative cultivars. I ask the authors to perform a phylogenetic tree using only a set of genes with the highest diversity. Also, a Table with the number of SNPs per gene/region would be informative (especially given that variation in size was highly conserved).

In addition, when comparing highly close relatives of the same species, ML or other cladistic approaches have been shown to be insufficient. You should perform another analysis like haplotype grouping that might be more informative/illustrative of your little variation in sequences.

Before this manuscript is ready for publication, I think that these analyses mentioned above should be conducted. Also, I have listed below minor errors that can be used to improve the manuscript. I have found several writing issues that need much attention.

Minor comments:

-The title has a spelling error (CharacteriZation)

-The term “elongated” to define the length of a gene is incorrect. Please rephrase along the text.

-Several grammar error are scattered along the text, like:

P1L33:… sliding window analysis and phylogenetic analysis were performed.

-Abstract: what do you mean with the last sentence? Please rephrase and clarify.

Introduction:

-Please check scale bars of Fig. 1 A-D. At least in A doesn’t seem to be 1cm.

-P2L65: I am not familiar with the term “scape”. I suggest to replace it with “inflorescence” or “shoot”.

-P2L108-109.Please rephrase the last sentence (bad use of English).

- 1.2.1 Problems WITH THEin nomenclature of cultivars. Anyway, I suggest to replace this subheading with something like: “Taxonomic status of Hyacinthus orientalis”; Nomenclatural circumscription of Hyacinthus orientalis”

-P4L135: replace “observed” with “characterized”.

-P4L158: “down” sounds strange to be used.

Materials and Methods:

-L177: inAT -80°C freezer

-L194: I am confused. Did you used genomic DNA for sequencing of cpDNA? This is one of the most importance piece of information for your work, and for that, authors should be clear about this particular point. In other words, how do you get rid of nuclear genomic DNA? If necessary for cpDNA sequencing at all.

-L211: alignmentS were, or, alignment wASere

-L213: correctedING

-L261: You have cases i, ii, and iii. Which one is 600 and 200bp?

-L271-274: the description is correct, but the use of English is very poor in these sentences.

L-381: no?????

Fig. 8. Frequency of LTRs in OF different length among the nine studied cpDNAs

L-394: indicate???? I think you added an extra word (no need it).

Fig. 11. Why the terminals of the tree have different colors? I didn’t find any reference to this in the caption. This must be described in the legend.

L486: basingED

L487: closest TO (not with)

Discussion:

-L513: length of nucleotides???? Perhaps, lenght of genes or regions, not nucleotides!!! A are always A, etc…

-L513-515: Indeed, you should mention the data used in Hu et al. I assume that they do not have the complete cpDNA as you.

- 4.3 Minor differences among the similar cpDNAs CHLOROPLAST GENOMES of hyacinth cultivars.

Author Response

Point 1: One of my concerns is the use of the color of flowers as a character to discuss the groups found. Given that several colors are unique, and for that, autapomorphic, this precludes its use for cladistic analysis. For grouping, you need by definition state characters that are shared by two or more terminals. Subheading 4.1 is conceptually incorrect. Also, the discussion about this point should be reconsidered. Moreover, is there any paper where the variation in flower color is consistent with cpDNA (at the infrageneric scale)?

Response: Thank you for your question. There is a paper of chloroplast genome studying Utricularia amethystina which has different flower colours and morphologies (Silva, 2019). Yet, this study did not show the consistency between flower color variation and the chloroplast genome. While another paper studied the phylogeny and phylogeography of Stellera chamaejasme (Zhang, 2010), using three chloroplast sequences (trnT-L, trnL-F and rpL16), found that the white-red flower is ancestral in the species, from which white-yellow and pure-red flowers are derived. 

The title of subsection 4.1 is changed as “Chloroplast genomes serve as new evidence in the phylogeny of Hyacinthus cultivars”, and the discussion is focused on the different phylogenetic positions of cultivars in our and previous studies.

Silva, S.R.; Pinheiro, D.G.; Penha, H.A.; PÅ‚achno, B.J.; Michael, T.P.; Meer, E.J.; Miranda, V.F.O.; Varani, A.M. Intraspecific Variation within the Utricularia amethystina Species Morphotypes Based on Chloroplast Genomes. Int J Mol Sci 2019, 20, 6130. https://doi.org/10.3390/ijms20246130

Zhang, Y.; Volis, S.; Sun, H. Chloroplast phylogeny and phylogeography of Stellera chamaejasme on the Qinghai-Tibet Plateau and in adjacent regions. Mol Phylogenet Evol 2010, 57, 1162-1172. https://doi.org/10.1016/j.ympev.2010.08.033

Point 2: Given the phylogenetic proximity of samples, and probably the domestication process underlying these cultivars, the diversity found was very slow. This is somehow expected for me, using cpDNA. The authors took a lot of effort into describing the lack of difference in genes and regions size, which for me is not promising from the beginning. Are other infrageneric studies within flowering plants where the size (insertions and deletions) are meaningful? For me, SNPs are always more promising. Although the phylogenetic tree is partially informative, there is a lack of resolution is in the Chicago and relative cultivars. I ask the authors to perform a phylogenetic tree using only a set of genes with the highest diversity. Also, a Table with the number of SNPs per gene/region would be informative (especially given that variation in size was highly conserved).

Response: Thank you for your suggestions. On top of the full alignment of the complete chloroplast genomes, the loci with Pi over 0.035—trnK-UUU-trnQ-UUG, trnS-GCU-trnG-UCC, petN-psbM, psbC, psbE-petL, ndhA, trnN-GUU-trnR-ACG, rrn23, trnA-UGC—were extracted and used for the construction of ML trees. Meanwhile, a full table of variable sites including SNPs and Indels is included in the manuscript as Table 5.

Point 3: In addition, when comparing highly close relatives of the same species, ML or other cladistic approaches have been shown to be insufficient. You should perform another analysis like haplotype grouping that might be more informative/illustrative of your little variation in sequences.

Response: Thank you for your suggestions. We performed haplotype grouping using DNaSP and generated a median joining network using Network 10. We present our methods and results in new subsections 2.6 Haplotype grouping and Identification of Molecular Diagnostic Characters and 3.6 Haplotype grouping and Identification of Molecular Diagnostic Characters, respectively. The results of haplotype grouping and identified MDCs were applied in the discussion 4.3 Differences among the chloroplast genomes of Hyacinthus cultivars.

Minor comments:

Point 4: -The title has a spelling error (CharacteriZation)

Response: We adopted British English throughout the manuscript. The British spelling for the American-spelled “characterization” is “characterisation”.

Point 5: -The term “elongated” to define the length of a gene is incorrect. Please rephrase along the text.

Response: We revised the term as “extended”.

Point 6: -Several grammar error are scattered along the text, like:

P1L33: … sliding window analysis and phylogenetic analysis were performed.

Response: Agreed. Please see the amendment in the manuscript.

Abstract:

Point 7: -What do you mean with the last sentence? Please rephrase and clarify.

Response: The last sentence in the abstract is revised as “Complete chloroplast genomes serve as new evidence in Hyacinthus cultivar phylogeny, contributing to cultivar identification, preservation and breeding.”

Introduction:

Point 8: -Please check scale bars of Fig. 1 A-D. At least in A doesn’t seem to be 1cm.

Response: Thank you for your reminding. Scale bar of Figure 1A was revised.

Point 9: -P2L65: I am not familiar with the term “scape”. I suggest to replace it with “inflorescence” or “shoot”.

Response: Thank you for your suggestions. “Scape” is a technical term in botanical description. According to The Cambridge Illustrated Glossary of Botanical Terms (Hickey & King, 2000), scape means “a leafless stalk, arising from the ground, which bears one or more single flowers, e.g. Hyacinthoides (bluebell), or a head of flowers e.g. Taraxacum (dandelion)”. In The Kew Plant Glossary - an illustrated dictionary of plant terms (2nd edition) (Beentjie, 2016), scape is “a leafless flower- or inflorescence-stalk arising from ground level, naked peduncle”. 

For easy understanding, the sentence is revised as following:

“The inflorescence of hyacinth is a raceme, with 2 to 40 flowers on a single scape [1-5] (leafless stalk arising from ground level [13,15]).”

Hickey, M. & King, C. (2000). The Cambridge Illustrated Glossary of Botanical Terms. Cambridge University Press, Cambridge. ISBN-13: 978-0-521-79401-5. ISBN-10: 0-521-79401-3.

Beentjie, H. (2016). The Kew Plant Glossary. 2nd edition, 1st in 2012. Kew Publishing, Royal Botanical Garden, Surrey, U.K. ISBN: 978-1-84246-604-9.

Point 10: -P2L108-109.Please rephrase the last sentence (bad use of English).

Response: We removed a portion of this sentence since the conveyed message duplicates.

Point 11: - 1.2.1 Problems WITH THEin nomenclature of cultivars. Anyway, I suggest to replace this subheading with something like: “Taxonomic status of Hyacinthus orientalis”; Nomenclatural circumscription of Hyacinthus orientalis”

Response: Agreed. We adopt the subheading “Nomenclatural circumscription of Hyacinthus cultivars with the emphasis on cultivars but not the species naturally exist in the wild.

Point 12: -P4L135: replace “observed” with “characterized”.

Response: Agreed. Please see the amendment in the manuscript.

Point 13: -P4L158: “down” sounds strange to be used.

Response: The words “down to” is replaced by “up to”.

Materials and Methods:

Point 14: -L177: inAT -80°C freezer

Response: Agreed. Please see the amendment in the manuscript.

Point 15: -L194: I am confused. Did you used genomic DNA for sequencing of cpDNA? This is one of the most importance piece of information for your work, and for that, authors should be clear about this particular point. In other words, how do you get rid of nuclear genomic DNA? If necessary for cpDNA sequencing at all.

Response: We extracted total genomic DNA for shotgun sequencing. The raw reads were paired up, trimmed, and de novo assembled into contigs. The contigs were mapped on a NC-verified reference chloroplast genome. During this step nuclear genomic DNA were excluded. The paragraph is revised as below.

“Genomic DNA of each specimen was extracted from about 50 mg of frozen fresh leaves using i-genomic Plant DNA Extraction Mini Kit (iNtRON Biotechnology, Daejeon, Korea) according to the instructions of the manufacturer. The concentration of extracted DNA was measured by NanoDrop Lite Spectrophotometer (Thermo Fisher Scientific, Massachusetts, USA), while the quality of DNA was checked by 1% agarose gel electrophoresis. The qualified genomic DNA were sent to Novogene Bioinformatic Technology Co. Ltd. (http://en.novogene.com/, Beijing, China) for shotgun sequencing.”

Also, to avoid confusion, all abbreviated “cpDNA(s)” is revised as “chloroplast genome(s)”.

Point 16: -L211: alignmentS were, or, alignment wASere

Response: Agreed. Revised as “alignment was”.

Point 17: -L213: correctedING

Response: Agreed. Please see the amendment in the manuscript.

Point 18: -L261: You have cases i, ii, and iii. Which one is 600 and 200bp?

Response: The sentence is rephrased as “The window length was set to 600 bp, and the step size was set to 200 bp.”.

Point 19: -L271-274: the description is correct, but the use of English is very poor in these sentences.

Response: The description is revised as the following:

“Maximum Likelihood (ML) tree was constructed using the software MEGA-X [68]. The best-fit nucleotide substitution model with the lowest Bayesian Information Crite-rion (BIC) scores was selected. The bootstrap replicates were set to 1,000. The full alignment of the fifteen complete chloroplast genomes, and the loci with Pi over 0.035—trnK-UUU-trnQ-UUG, trnS-GCU-trnG-UCC, petN-psbM, psbC, psbE-petL, ndhA, trnN-GUU-trnR-ACG, rrn23 and trnA-UGC—were used for the construction of ML trees.”

Point 20: -L-381: no?????

Reponse: “having no” is replaced by “without”.

Point 21: Fig. 8. Frequency of LTRs in OF different length among the nine studied cpDNAs

Reponse: Agreed. Please see the amendment in the manuscript.

Point 22: L-394: indicate???? I think you added an extra word (no need it).

Response: Sorry for the typo. The word “indicate” before the full stop is removed. Figure 9 was moved to supplementary documents. The caption is attached with the figure in Figure S4.

Point 23: Fig. 11. Why the terminals of the tree have different colors? I didn’t find any reference to this in the caption. This must be described in the legend.

Response: For the phylogenetic trees in Figure 10, the sentence “The accession numbers and species names were coloured according to flower colours (white is presented as grey instead)” is added.

Point 24: L486: basingED

Response: Agreed. But the sentence is removed since the description of results is modified.

Point 25: L487: closest TO (not with)

Response: Agreed. Please see the amendment in the manuscript.

Discussion:

Point 26: -L513: length of nucleotides???? Perhaps, lenght of genes or regions, not nucleotides!!! A are always A, etc…

Response: Thank you so much for your clarification! We agreed but the sentence is removed since the discussion of subsection 4.1 is modified.

Point 27: -L513-515: Indeed, you should mention the data used in Hu et al. I assume that they do not have the complete cpDNA as you.

Response: The discussion of subsection 4.1 is revised, and the first paragraph recalled the work done by Hu et al.

Point 28: - 4.3 Minor differences among the similar cpDNAs CHLOROPLAST GENOMES of hyacinth cultivars.

Response: Agreed. The title of subsection 4.1 is revised as “Differences among the chloroplast genomes of Hyacinthus cultivars”.

Round 2

Reviewer 1 Report

The revised version is significantly improved, I recommend it for publication

Reviewer 2 Report

After reading the revised version of the manuscript presented by Wong and collaborators and their author's reply, now I consider that this work is ready for publication. 

The authors have addressed all my comments and the manuscript was largerly improved.